# REVIVING AUTOENCODER PRETRAINING

## ABSTRACT

The pressing need for pretraining algorithms has been diminished by numerous advances in terms of regularization, architectures, and optimizers. Despite this trend, we re-visit the classic idea of unsupervised autoencoder pretraining and propose a modified variant that relies on a full reverse pass trained in conjunction with a given training task. We establish links between SVD and pretraining and show how it can be leveraged for gaining insights about the learned structures. Most importantly, we demonstrate that our approach yields an improved performance for a wide variety of relevant learning and transfer tasks ranging from fully connected networks over ResNets to GANs. Our results demonstrate that unsupervised pretraining has not lost its practical relevance in today's deep learning environment.

## 1 INTRODUCTION

While approaches such as greedy layer-wise autoencoder pretraining (Bengio et al., 2007; Vincent et al., 2010; Erhan et al., 2010) arguably paved the way for many fundamental concepts of today's methodologies in deep learning, the pressing need for pretraining neural networks has been diminished in recent years. This was primarily caused by numerous advances in terms of regularization (Srivastava et al., 2014; Hanson & Pratt, 1989; Weigend et al., 1991), network architectures (Ronneberger et al., 2015; He et al., 2016; Vaswani et al., 2017), and improved optimization algorithms (Kingma & Ba, 2014; Loshchilov & Hutter, 2017; Reddi et al., 2019). Despite these advances, training deep neural networks that generalize well to a wide range of previously unseen tasks remains a fundamental challenge (Neyshabur et al., 2017; Kawaguchi et al., 2017; Frankle & Carbin, 2018).

Inspired by techniques for orthogonalization (Ozay & Okatani, 2016; Jia et al., 2017; Bansal et al., 2018), we re-visit the classic idea of unsupervised autoencoder pretraining in the context of reversible network architectures. Hence, we propose a modified variant that relies on a full reverse

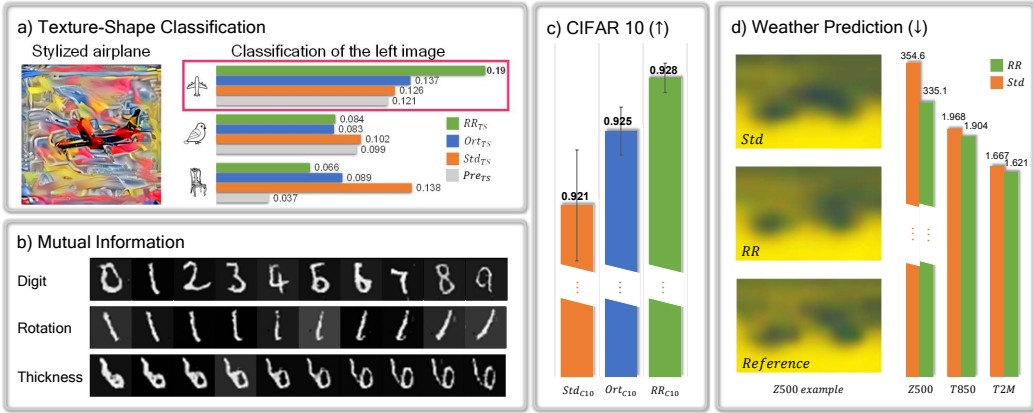

Figure 1: Our pretraining (denoted as RR) yields improvements for numerous applications: **a)**: For difficult shape classification tasks, it outperforms existing approaches (Std$_{TS}$, Ort$_{TS}$, Pre$_{TS}$): the RR$_{TS}$ model classifies the airplane shape with significantly higher confidence. **b)**: Our approach establishes mutual information between input and output distributions. **c)**: For CIFAR 10 classification with a Resnet110, RR$_{C10}$ yields substantial practical improvements over the state-of-the-art. **d)**: Learned weather forecasting has strictly limited real-world data: our pretraining yields improvements for pressure (Z500, zoomed in regions shown above), atmospheric temperature (T850) as well as ground temperature (T2M).

pass trained in conjunction with a given training task. A key insight is that there is no need for "greediness", i.e., layer-wise decompositions of the network structure, and it is additionally beneficial to take into account a specific problem domain at the time of pretraining. We establish links between singular value decomposition (SVD) and pretraining, and show how our approach yields an embedding of problem-aware dominant features in the weight matrices. An SVD can then be leveraged to conveniently gain insights about learned structures. Most importantly, we demonstrate that the proposed pretraining yields an improved performance for a variety of learning and transfer tasks. Our formulation incurs only a very moderate computational cost, is very easy to integrate, and widely applicable.

The structure of our networks is influenced by invertible network architectures that have received significant attention in recent years (Gomez et al., 2017; Jacobsen et al., 2018; Zhang et al., 2018a). However, instead of aiming for a bijective mapping that reproduces inputs, we strive for learning a general representation by constraining the network to represent an as-reversible-as-possible process for all *intermediate* layer activations. Thus, even for cases where a classifier can, e.g., rely on color for inference of an object type, the model is encouraged to learn a representation that can recover the input. Hence, not only the color of the input should be retrieved, but also, e.g., its shape. In contrast to most structures for invertible networks, our approach does not impose architectural restrictions. We demonstrate the benefits of our pretraining for a variety of architectures, from fully connected layers to convolutional neural networks (CNNs), over networks with and without batch normalization, to GAN architectures. We discuss other existing approaches and relate them to the proposed method in the appendix.

Below, we will first give an overview of our formulation and its connection to singular values, before evaluating our model in the context of transfer learning. For a regular, i.e., a non-transfer task, the goal usually is to train a network that gives optimal performance for one specific goal. During a regular training run, the network naturally exploits any observed correlations between input and output distribution. An inherent difficulty in this setting is that typically no knowledge about the specifics of the new data and task domains is available when training the source model. Hence, it is common practice to target broad and difficult tasks hoping that this will result in features that are applicable in new domains (Zamir et al., 2018; Gopalakrishnan et al., 2017; Ding et al., 2017). Motivated by autoencoder pretraining, we instead leverage a pretraining approach that takes into account the data distribution of the inputs. We demonstrate the gains in accuracy for original and new tasks below for a wide range of applications, from image classification to data-driven weather forecasting.

## 2 METHOD

With state-of-the-art methods, there is no need for breaking down the training process into single layers. Hence, we consider approaches that target whole networks, and especially orthogonalization regularizers as a starting point (Huang et al., 2018). Orthogonality constraints were shown to yield improved training performance in various settings (Bansal et al., 2018), and can be formulated as:

$$\mathcal{L}_{\text{ort}} = \sum_{m=1}^{n} \left\| M_m^T M_m - I \right\|_F^2, \tag{1}$$

i.e., enforcing the transpose of the weight matrix $M_m \in \mathbb{R}^{s_m^{\text{out}} \times s_m^{\text{in}}}$ for all layers $m$ to yield its inverse when being multiplied with the original matrix. $I$ denotes the identity matrix with $I = (\mathbf{e}_m^1, ... \mathbf{e}_m^{s_m^{\text{in}}})$, $\mathbf{e}_m^j$ denoting the $j_{th}$ column unit vector. Minimizing equation 1, i.e. $M_m^T M_m - I = 0$ is mathematically equivalent to:

$$M_m^T M_m \mathbf{e}_m^j - \mathbf{e}_m^j = \mathbf{0}, j = 1, 2, ..., s_m^{\text{in}}, \tag{2}$$

with $rank(M_m^T M_m) = s_m^{\text{in}}$, and $\mathbf{e}_m^j$ as eigenvectors of $M_m^T M_m$ with eigenvalues of 1. This formulation highlights that equation 2 does not depend on the training data, and instead only targets the content of $M_m$. Inspired by the classical unsupervised pretraining, we re-formulate the orthogonality constraint in a *data-driven* manner to take into account the set of inputs $\mathcal{D}_m$ for the current layer (either activation from a previous layer or the training data $\mathcal{D}_1$), and instead minimize

$$\mathcal{L}_{\text{RR}} = \sum_{m=1}^{n} (M_m^T M_m \mathbf{d}_m^i - \mathbf{d}_m^i)^2 = \sum_{m=1}^{n} ((M_m^T M_m - I) \mathbf{d}_m^i)^2, \tag{3}$$

where $\mathbf{d}_m^i \in \mathcal{D}_m \subset \mathbb{R}^{s_m^{\text{in}}}$. Due to its reversible nature, we will denote our approach with an RR subscript in the following. In contrast to classical autoencoder pretraining, we are minimizing this loss jointly for all layers of a network, and while orthogonality only focuses on $M_m$, our formulation allows for minimizing the loss by extracting the dominant features of the input data.

Let $q$ denote the number of linearly independent entries in $\mathcal{D}_m$, i.e. its dimension, and $t$ the size of the training data, i.e. $|\mathcal{D}_m| = t$, usually with $q < t$. For every single datum $\mathbf{d}_m^i, i = 1, 2, ..., t$, equation 3 results in

$$M_m^T M_m \mathbf{d}_m^i - \mathbf{d}_m^i = \mathbf{0}, \tag{4}$$

and hence $\mathbf{d}_m^i$ are eigenvectors of $M_m^T M_m$ with corresponding eigenvalues being 1. Thus, instead of the generic constraint $M_m^T M_m = I$ that is completely agnostic to the data at hand, the proposed formulation of equation 4 is aware of the training data, which improves the generality of the learned representation, as we will demonstrate in detail below.

As by construction, $rank(M_m) = r \leqslant min(s_m^{\text{in}}, s_m^{\text{out}})$, the SVD of $M_m$ yields:

$$M_m = U_m \Sigma_m V_m^T, \text{ with} \begin{cases} U_m = (\mathbf{u}_m^1, \mathbf{u}_m^2, ..., \mathbf{u}_m^r, \mathbf{u}_m^{r+1}, ..., \mathbf{u}_m^{s_m^{\text{out}}}) \in \mathbb{R}^{s_m^{\text{out}} \times s_m^{\text{out}}}, \\ V_m = (\mathbf{v}_m^1, \mathbf{v}_m^2, ..., \mathbf{v}_m^r, \mathbf{v}_m^{r+1}, ..., \mathbf{v}_m^{s_m^{\text{in}}}) \in \mathbb{R}^{s_m^{\text{in}} \times s_m^{\text{in}}}, \end{cases} \tag{5}$$

with left and right singular vectors in $U_m$ and $V_m$, respectively, and $\Sigma_m$ having square roots of the $r$ eigenvalues of $M_m^T M_m$ on its diagonal. $\mathbf{u}_m^k$ and $\mathbf{v}_m^k (k = 1, ..., r)$ are the eigenvectors of $M_m M_m^T$ and $M_m^T M_m$, respectively (Wall et al., 2003). Here, especially the right singular vectors in $V_m^T$ are important, as they determine which structures of the input are processed by the transformation $M_m$. The original orthogonality constraint with equation 2 yields $r$ unit vectors $\mathbf{e}_m^j$ as the eigenvectors of $M_m^T M_m$. Hence, the influence of equation 2 on $V_m$ is completely independent of training data and learning objectives.

Next, we show that $\mathcal{L}_{\text{RR}}$ facilitates learning dominant features from a given data set. For this, we consider an arbitrary basis for spanning the space of inputs $\mathcal{D}_m$ for layer $m$. Let $\mathcal{B}_m : \langle \mathbf{w}_m^1, ..., \mathbf{w}_m^q \rangle$ denote a set of $q$ orthonormal basis vectors obtained via a Gram-Schmidt process, with $t \geqslant q \geqslant r$, and $D_m$ denoting the matrix of the vectors in $\mathcal{B}_m$. As we show in more detail in the appendix, our constraint from equation 4 requires eigenvectors of $M_m^T M_m$ to be $\mathbf{w}_m^i$, with $V_m$ containing $r$ orthogonal vectors $(\mathbf{v}_m^1, \mathbf{v}_m^2, ..., \mathbf{v}_m^r)$ from $\mathcal{D}_m$ and $(s_m^{\text{in}} - r)$ vectors from the null space of $M$.

We are especially interested in how $M_m$ changes w.r.t. input in terms of $D_m$, i.e., we express $\mathcal{L}_{\text{RR}}$ in terms of $D_m$. By construction, each input $\mathbf{d}_m^i$ can be represented as a linear combination via a vector of coefficients $\mathbf{c}_m^i$ that multiplies $D_m$ so that $\mathbf{d}_m^i = D_m \mathbf{c}_m^i$. Since $M_m \mathbf{d}_m = U_m \Sigma_m V_m^T \mathbf{d}_m$, the loss $\mathcal{L}_{\text{RR}}$ of layer $m$ can be rewritten as

$$\begin{aligned} \mathcal{L}_{\text{RR}_m} &= (M_m^T M_m \mathbf{d}_m - \mathbf{d}_m)^2 = (V_m \Sigma_m^T \Sigma_m V_m^T \mathbf{d}_m - \mathbf{d}_m)^2 \\ &= (V_m \Sigma_m^T \Sigma_m V_m^T D_m \mathbf{c}_m - D_m \mathbf{c}_m)^2, \end{aligned} \tag{6}$$

where we can assume that the coefficient vector $\mathbf{c}_m$ is accumulated over the training data set size $t$ via $\mathbf{c}_m = \sum_{i=1}^t \mathbf{c}_m^i$, since eventually every single datum in $\mathcal{D}_m$ will contribute to $\mathcal{L}_{\text{RR}_m}$. The central component of equation 6 is $V_m^T D_m$. For a successful minimization, $V_m$ needs to retain those $\mathbf{w}_m^i$ with the largest $\mathbf{c}_m$ coefficients. As $V_m$ is typically severely limited in terms of its representational capabilities by the number of adjustable weights in a network, it needs to focus on the most important eigenvectors in terms of $\mathbf{c}_m$ in order to establish a small distance to $D_m \mathbf{c}_m$. Thus, features that appear multiple times in the input data with a corresponding factor in $\mathbf{c}_m$ will more strongly contribute to minimizing $\mathcal{L}_{\text{RR}_m}$.

To summarize, $V_m$ is driven towards containing $r$ orthogonal vectors $\mathbf{w}_m^i$ that represent the most frequent features of the input data, i.e., the dominant features. Additionally, due to the column vectors of $V_m$ being mutually orthogonal, $M_m$ is encouraged to extract different features from the input. By the sake of being distinct and representative for the data set, these features have the potential to be useful for new inference tasks. The feature vectors embedded in $M_m$ can be extracted from the network weights in practical settings, as we will demonstrate below.

**Realization in Neural Networks**  Calculating $M_m^T M_m$ is usually very expensive due to the dimensionality of $M_m$. Instead of building it explicitly, we constrain intermediate results to realize equation 3 when training. A regular training typically starts with a chosen network structure and

trains the model weights for a given task via a suitable loss function. Our approach fully retains this setup and adds a second pass that reverses the initial structure while reusing all weights and biases. E.g., for a typical fully connected layer in the forward pass with $\mathbf{d}_{m+1} = M_m\mathbf{d}_m + \mathbf{b}_m$, the reverse pass operation is given by $\mathbf{d}'_m = M_m^T(\mathbf{d}_{m+1} - \mathbf{b}_m)$, where $\mathbf{d}'_m$ denotes the reconstructed input.

Our goal with the reverse pass is to transpose all operations of the forward pass to obtain identical intermediate activations between the layers with matching dimensionality. We can then constrain the intermediate results of each layer of the forward pass to match the results of the backward pass, as illustrated in figure 2. Unlike greedy layer-wise autoencoder pretraining, which trains each layer separately and only constrains $\mathbf{d}_1$ and $\mathbf{d}'_1$, we jointly train all layers and constrain all intermediate results. Due to the symmetric structure of the two passes, we can use a simple $L^2$ difference to drive the network towards aligning the results:

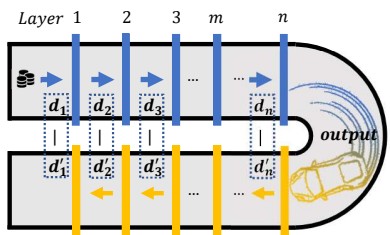

Figure 2: A visual overview of the regular forward pass (blue) and the corresponding reverse pass for pretraining (yellow).

$$\mathcal{L}_{\text{RR}} = \sum_{m=1}^{n} \lambda_m \left\| \mathbf{d}_m - \mathbf{d}'_m \right\|_F^2. \tag{7}$$

Here $\mathbf{d}_m$ denotes the input of layer $m$ in the forward pass and $\mathbf{d}'_m$ the output of layer $m$ for the reverse pass. $\lambda_m$ denotes a scaling factor for the loss of layer $m$, which, however, is typically constant in our tests across all layers. Note that with our notation, $\mathbf{d}_1$ and $\mathbf{d}'_1$ refer to the input data, and the reconstructed input, respectively.

Next, we show how this setup realizes the regularization from equation 3. For clarity, we use a fully connected layer with bias. In a neural network with $n$ hidden layers, the forward process for a layer $m$ is given by $\mathbf{d}_{m+1} = M_m\mathbf{d}_m + \mathbf{b}_m$,, with $\mathbf{d}_1$ and $\mathbf{d}_{n+1}$ denoting in- and output, respectively. For our pretraining, we build a reverse pass network with transposed operations starting with the final output where $\mathbf{d}_{n+1} = \mathbf{d}'_{n+1}$, and the intermediate results $\mathbf{d}'_{m+1}$:

$$\mathbf{d}'_m = M_m^T(\mathbf{d}'_{m+1} - \mathbf{b}_m), \tag{8}$$

which yields $\left\| \mathbf{d}_m - \mathbf{d}'_m \right\|_F^2 = \left\| M_m^T M_m \mathbf{d}_m - \mathbf{d}_m \right\|_F^2$. When this difference is minimized via equation 7, we obtain activated intermediate content during the reverse pass that reconstructs the values computed in the forward pass, i.e. $\mathbf{d}'_{m+1} = \mathbf{d}_{m+1}$ holds. As in equation 10 the reverse pass activation $\mathbf{d}'_m$ depends on $\mathbf{d}_{m+1}'$, this formulation yields a full reverse pass from output to input, which we use for most training runs below. In this case

$$\mathbf{d}'_m = M_m^T(\mathbf{d}'_{m+1} - \mathbf{b}_m) = M_m^T(\mathbf{d}_{m+1} - \mathbf{b}_m) = M_m^T M_m \mathbf{d}_m , \tag{9}$$

which is consistent with equation 3, and satisfies the original constraint $M_m^T M_m \mathbf{d}_m - \mathbf{d}_m = \mathbf{0}$. This version is preferable if a unique path from output to input exists. For architectures where the path is not unique, e.g., in the presence of additive residual connections, we use a local formulation

$$\mathbf{d}'_m = M_m^T(\mathbf{d}_{m+1} - \mathbf{b}_m), \tag{10}$$

which employs $\mathbf{d}_{m+1}$ for jointly constraining all intermediate activations in the reverse pass.

Up to now, the discussion focused on simplified neural networks without activation functions or extensions such as batch normalization (BN). While we leave incorporating such extensions for future work, our experiments consistently show that the inherent properties of our pretraining remain valid: even with activations and BN, our approach successfully extracts dominant structures and yields improved generalization. In the appendix, we give details on how to ensure that the latent space content for forward and reverse pass is aligned such that differences can be minimized.

To summarize, we realize the loss formulation of equation 7 to minimize $\sum_{m=1}^{n}((M_m^T M_m - I)\mathbf{d}_m)^2$ without explicitly having to construct $M_m^T M_m$. Following the notation above, we will refer to networks trained with the added reverse structure and the additional loss terms as *RR* variants. We consider two variants for the reverse pass: a local pretraining equation 10 using the datum $\mathbf{d}_{m+1}$ of a given layer, and a full version via equation 8 which uses $\mathbf{d}'_{m+1}$ incoming from the next layer during the reverse pass.

**Embedding Singular Values** Below, Std denotes a regular training run (in orange color in graphs below), while RR denotes our models (in green). Pre and Ort will denote regular autoencoder pretraining and orthogonality, respectively, while a subscript will denote the task variant the model was trained for, e.g., $Std_T$ for task $T$. While we typically use all layers of a network in the constraints, a reduced variant that we compare to below only applies the constraint for the input data, i.e., m=1. A network trained with this variant, denoted by $RR_A^1$, is effectively trained to only reconstruct the input. It contains no constraints for the inner activations and layers of the network. For the Ort models, we use the Spectral Restricted Isometry Property algorithm (Bansal et al., 2018).

We verify that the column vectors of $V_m$ of models from RR training contain the dominant features of the input with the help of a classification test, employing a single fully connected layer, i.e., $\mathbf{d}_2 = M_1 \mathbf{d}_1$, with batch normalization and activation. To quantify this similarity, we compute an LPIPS distance (Zhang et al., 2018b) between $v_m^i$ and the training data (lower values being better).

We employ a training data set constructed from two dominant classes (a peak in the top left, and bottom right quadrant, respectively), augmented with noise in the form of random scribbles. Based on the analysis above, we expect the RR training to extract the two dominant peaks during training. The LPIPS measurements confirm our SVD argumentation above, with average scores of $0.217 \pm 0.022$ for RR, $0.319 \pm 0.114$ for Pre, $0.495 \pm 0.006$ for Ort, and $0.500 \pm 0.002$ for Std. I.e., the RR model fares significantly better than the others. At the same time, the peaks are clearly visible for RR models, an example is shown in figure 3(b), while the other models fail to extract structures that resemble the input. Thus, by training with the full network and the original training objective, our pretraining yields structures that are interpretable and be inspected by humans.

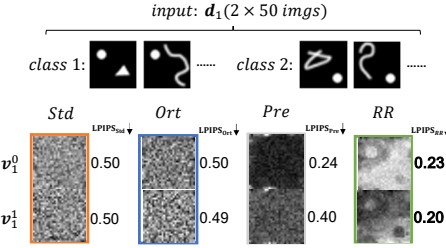

Figure 3: Column vectors of $V_m$ for different trained models Std, Ort, Pre and RR for peaks. Input features clearly are successfully embedded in the weights of RR, as confirmed by the LPIPS scores.

The results above experimentally confirm our formulation of the RR loss and its ability to extract dominant and generalizing structures from the training data. Next, we will focus on quantified metrics and turn to measurements in terms of mutual information to illustrate the behavior of our pretraining for deeper networks.

## 3 EVALUATION IN TERMS OF MUTUAL INFORMATION

As our approach hinges on the introduction of the reverse pass, we will show that it succeeds in terms of establishing mutual information (MI) between the input and the constrained intermediates inside a network. More formally, MI $I(X; Y)$ of random variables $X$ and $Y$ measures how different the joint distribution of $X$ and $Y$ is w.r.t. the product of their marginal distributions, i.e., the Kullback-Leibler divergence $I(X; Y) = D_{KL}[P_{(X,Y)} || P_X P_Y]$. (Tishby & Zaslavsky, 2015) proposed *MI plane* to analyze trained models, which show the MI between the input $X$ and activations of a layer $\mathcal{D}_m$, i.e., $I(X; \mathcal{D}_m)$ and $I(\mathcal{D}_m; Y)$, i.e., MI of layer $\mathcal{D}_m$ with output $Y$. These two quantities indicate how much information about the in- and output distributions are retained at each layer, and we use them to show to which extent our pretraining succeeds at incorporating information about the inputs throughout training.

The following tests employ networks with six fully connected layers with the objective to learn the mapping from 12 binary inputs to 2 binary output digits (Shwartz-Ziv & Tishby, 2017), with results accumulated over five runs. We compare the versions $Std_A$, $Pre_A$, $Ort_A$, $RR_A$, and a variant of the latter: $RR_A^1$, i.e. a version where only the input $\mathbf{d}_1$ is constrained to be reconstructed. While figure 4a) visually summarizes the content of the MI planes, the graph in (b) highlights that training with the RR loss correlates input and output distributions across all layers: the cluster of green points in the center of the graph shows that all layers contain balanced MI between in- as well as output and the activations of each layer. $RR_A^1$ fares slightly worse, while $Std_A$ and $Ort_A$ almost exclusively focus on the output with $I(\mathcal{D}_m; Y)$ being close to one. $Pre_A$ instead only focuses on reconstructing inputs. Thus, the early layers cluster in the right-top corner, while the last layer $I(\mathcal{D}_7; Y)$ fails to

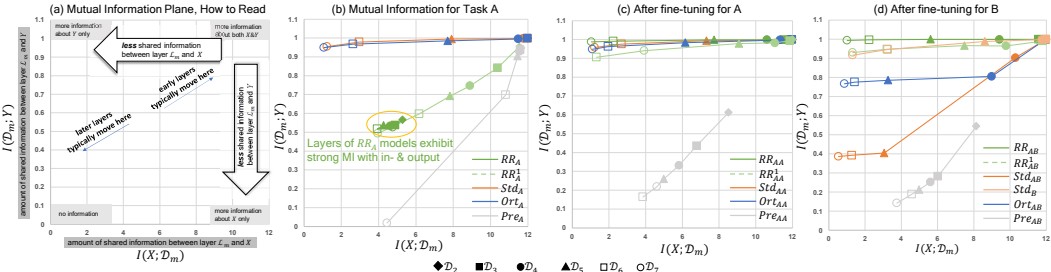

Figure 4: MI planes for different models: a) Visual overview of the contents. b) Plane for task A. Points on each line correspond to layers of one type of model. All points of $RR_A$, are located in the center of the graph, while $Std_A$ and $Ort_A$, exhibit large $I(\mathcal{D}_m; Y)$, i.e., specialize on the output. $Pre_A$ strongly focuses on reconstructing the input with high $I(X; \mathcal{D}_m)$ for early layers. c,d): After fine-tuning for A/B. The last layer $\mathcal{D}_7$ of $RR_{AA}$ and $RR_{AB}$ successfully builds the strongest relationship with $Y$, yielding the highest accuracy.

align with the outputs. Once we continue fine-tuning these models without regularization, the MI naturally shifts towards the output, as shown in figure 4 (c). Here, $RR_{AA}$ outperforms the other models in terms of final performance. Likewise, $RR_{AB}$ performs best for a transfer task B with switched output digits, as shown in graph (d). The final performance for both tasks across all runs is summarized in figure 5. These graphs visualize that the proposed pretraining succeeds in robustly establishing mutual information between inputs and targets across a full network, in addition to extracting reusable features.

MI has received attention recently as a learning ob-
jective, e.g., in the form of the InfoGAN approach
(Chen et al., 2016) for learning disentangled and in-
terpretable latent representations. While MI is typ-
ically challenging to assess and estimate (Walters-
Williams & Li, 2009), the results above show that
our approach provides a straightforward and robust
way for including it as a learning objective. In this
way, we can, e.g., reproduce the disentangling re-
sults from (Chen et al., 2016), which are shown in
figure 1(c). A generative model with our pretraining
extracts intuitive latent dimensions for the different
digits, line thickness, and orientation without any
additional modifications of the loss function. The
joint training of the full network with the proposed
reverse structure, including non-linearities and nor-
malization, yields a natural and intuitive decomposi-
tion.

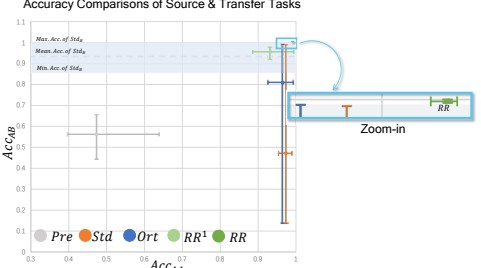

Figure 5: Performance for MI source and trans-
fer tasks for the models of figure 4. Due to the
large standard deviation of $Ort$, we show min/max
value ranges. The dashed gray line and region
show baseline accuracy for $Std_B$. The top-left in-
set highlights the stability of the high accuracy re-
sults from RR training.

## 4 EXPERIMENTAL RESULTS

We now turn to a broad range of network structures, i.e., CNNs, Autoencoders, and GANs, with a variety of data sets and tasks to show our approach succeeds in improving inference accuracy and generality for modern day applications and architectures.

**Transfer-learning Benchmarks** We first evaluate our approach with two state-of-the-art bench-
marks for transfer learning. The first one uses the *texture-shape* data set from (Geirhos et al., 2018), which contains challenging images of various shapes combined with patterns and textures to be classified. The results below are given for 10 runs each. For the stylized data shown in figure 6 (a), the accuracy of $Pre_{TS}$ is low with 20.8%. This result is in line with observations in previous work and confirms the detrimental effect of classical pretraining. $Std_{TS}$ yields a performance of 44.2%, and $Ort_{TS}$ improves the performance to 47.0%, while $RR_{TS}$ yields a performance of 54.7% (see figure 6b). Thus, the accuracy of $RR_{TS}$ is 162.98% higher than $Pre_{TS}$, 23.76% higher than $Std_{TS}$, and 16.38% higher than $Ort_{TS}$. To assess generality, we also apply the models to new data without

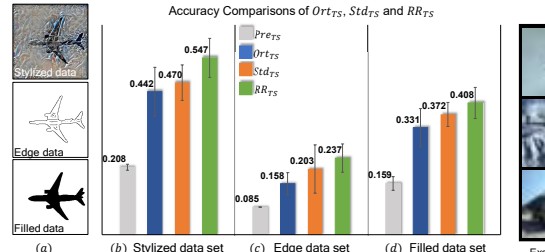
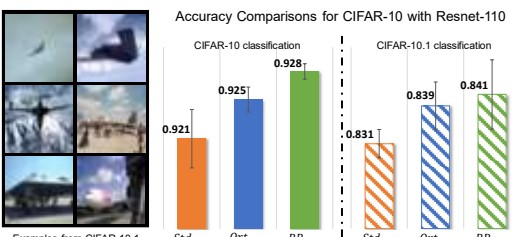

Figure 6: (a) Examples from texture-shape data set. (b, c, d) Texture-shape test accuracy comparisons of Pre_TS, Ort_TS, Std_TS and RR_TS for different data sets.

Figure 7: Left: Examples from CIFAR 10.1 data set. Right: Accuracy comparisons when applying models trained on CIFAR 10 to CIFAR 10.1 data.

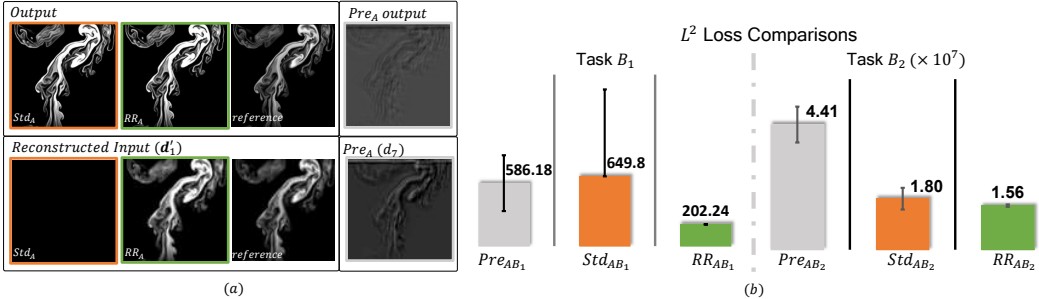

Figure 8: (a) Example output and reconstructed inputs, with the reference shown right. Only RR_A successfully recovers the input, Std_A produces a black image, while Pre_A fares poorly. (b) $L^2$ loss comparisons for two different generative transfer learning tasks (averaged across 5 runs each). The RR models show the best performance for both tasks.

re-training, i.e. an edge and a filled data set, also shown in figure 6 (a). For the edge data set, RR_TS outperforms Pre_TS, Std_TS and Ort_TS by 178.82%, 50% and 16.75%, respectively,

It is worth pointing out that the additional constraints of our training approach lead to moderately increased requirements for memory and computations, e.g., 41.86% more time per epoch than regular training for the texture-shape test. On the other hand, it allows us to train smaller models: we can reduce the weight count by 32% for the texture-shape case while still being on-par with Ort_TS in terms of classification performance. By comparison, regular layer-wise pretraining requires a significant overhead and fundamental changes to the training process. Our pretraining fully integrates with existing training methodologies and can easily be deactivated via $\lambda_m = 0$.

As a second test case, we use a CIFAR-based task transfer (Recht et al., 2019) that measures how well models trained on the original CIFAR 10, generalize to a new data set (CIFAR 10.1) collected according to the same principles as the original one. Here we use a Resnet110 with 110 layers and 1.7 million parameters, Due to the consistently low performance of the Pre models (Alberti et al., 2017), we focus on Std, Ort and RR for this test case. In terms of accuracy across 5 runs, Ort_C10 outperforms Std_C10 by 0.39%, while RR_C10 outperforms Ort_C10 by another 0.28% in terms of absolute test accuracy (figure 7). This increase for RR training matches the gains reported for orthogonality in previous work (Bansal et al., 2018), thus showing that our approach yields substantial practical improvements over the latter. It is especially interesting how well performance for CIFAR 10 translates into transfer performance for CIFAR 10.1. Here, RR_C10 still outperforms Ort_C10 and Std_C10 by 0.22% and 0.95%, respectively. Hence, the models from our pretraining very successfully translate gains in performance from the original task to the new one, indicating that the models have successfully learned a set of more general features. To summarize, both benchmark cases confirm that the proposed pretraining benefits generalization.

**Generative Adversarial Models**   In this section, we employ our pretraining in the context of generative models for transferring from synthetic to real-world data from the ScalarFlow data set (Eckert et al., 2019). As super-resolution task $A$, we first use a fully-convolutional generator network, adversarially trained with a discriminator network on the synthetic flow data. While regular pretraining is

(a) Three prediction examples of Z500    (b) Three prediction examples of T850    (c) Three prediction examples of T2M

Figure 9: Details of the three physical quantities of the weather forecasting test (full frames are shown in the appendix). As confirmed by the quantified results, RR predicts results closer to the reference.

more amenable to generative tasks than orthogonal regularization, it can not be directly combined with adversarial training. Hence, we pretrain a model Pre for a reconstruction task at high-resolution without discriminator instead. Figure 8 a) demonstrates that our method works well in conjunction with the GAN training: As shown in the bottom row, the trained generator succeeds in recovering the input via the reverse pass without modifications. A regular model $Std_A$, only yields a black image in this case. For $Pre_A$, the layer-wise nature of the pretraining severely limits its capabilities to learn the correct data distribution (Zhou et al., 2014), leading to a low performance.

We now mirror the generator model from the previous task to evaluate an autoencoder structure that we apply to two different data sets: the synthetic smoke data used for the GAN training (task $B_1$), and a real-world RGB data set of smoke clouds (task $B_2$). Thus both variants represent transfer tasks, the second one being more difficult due to the changed data distribution. The resulting losses, summarized in figure 8 b), show that RR training performs best for both autoencoder tasks: the $L^2$ loss of $RR_{AB_1}$ is 68.88% lower than $Std_{AB_1}$, while it is 13.3% lower for task $B_2$. The proposed pretraining also clearly outperforms the Pre variants. Within this series of tests, the RR performance for task B2 is especially encouraging, as this task represents a synthetic to real transfer.

**Weather Forecasting**    Pretraining is particularly attractive in situations where the amount of data for training is severely limited. Weather forecasting is such a case, as systematic and accurate data for many relevant quantities are only available for approximately 50 years. We target three-day forecasts of pressure, ground temperature, and mid-level atmospheric temperature based on a public benchmark dataset (Rasp et al., 2020). This dataset contains worldwide observations from ERA5 (Hersbach et al., 2020) in six-hour intervals with a 5.625° resolution. For the joint inference of atmospheric pressure (500 hPa geopotential, Z500), ground temperature (T2M), and atmospheric temperature (at 850 hPa, T850), we use a convolutional ResNet architecture with 19 residual blocks. As regular pretraining is not compatible with residual connections, we omit it here.

We train a model regular model (about 6.36M trainable parameters) with data from 1979 to 2015, and compare its inference accuracy across all datapoints from years 2017 and 2018 to a similar model that employs our pretraining. While the regular model was trained for 25 epochs, the RR model was pretrained for 10 epochs and fine-tuned for another 15 epochs. Across all three physical quantities, the RR model clearly outperforms the regular model, as summarized in figure 1 (d) and figure 9 (details are given in the appendix). Especially for the latitude-weighted RMSE of Z500, it yields improvements of 5.5%. These improvements point to an improved generalization of the RR model via the pretraining and highlight its importance for domains where data is scarce.

## 5    Conclusions

We have proposed a novel pretraining approach inspired by classic methods for unsupervised autoencoder pretraining and orthogonality constraints. In contrast to the classical methods, we employ a constrained reverse pass for the full non-linear network structure and include the original learning objective. We have shown for a wide range of scenarios, from mutual information, over transfer learning benchmarks to weather forecasting, that the proposed pretraining yields networks with better generalizing capabilities. Our training approach is general, easy to integrate, and imposes no requirements regarding network structure or training methods. Most importantly, our results show that unsupervised pretraining has not lost its relevance in today's deep learning environment.

As future work, we believe it will be exciting to evaluate our approach in additional contexts, e.g., for temporal predictions (Hochreiter & Schmidhuber, 1997; Cho et al., 2014), and for training explainable and interpretable models (Zeiler & Fergus, 2014; Chen et al., 2016; Du et al., 2018).

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

# A APPENDIX

To ensure reproducibility, source code and data for all tests will be published. Runtimes were measured on a machine with Nvidia GeForce GTX 1080 Ti GPUs and an Intel Core i7-6850K CPU.

## A.1 DISCUSSION OF RELATED WORK

Greedy layer-wise pretraining was first proposed by (Bengio et al., 2007), and influenced a large number of follow up works, providing a crucial method for enabling stable training runs of deeper networks. A detailed evaluation was performed by (Erhan et al., 2010), also highlighting cases were it can be detrimental. These problems were later on also detailed in other works, e.g., by (Alberti et al., 2017). The transferability of learned features was likewise a topic of interest for transfer learning applications (Yosinski et al., 2014). Sharing similarities with our approach, (Rasmus et al., 2015) combined supervised and unsupervised learning objectives, but focused on denoising autoencoders and a layer-wise approach without weight sharing. We demonstrate the importance of leveraging state-of-the-art methods for training deep networks, i.e. without decomposing or modifying the network structure. This not only improves performance, but also very significantly simplifies the adoption of the pretraining pass in new application settings.

Extending the classic viewpoint of unsupervised autoencoder pretraining, several prior methods employed "hard orthogonal constraints" to improve weight orthogonality via singular value decomposition (SVD) at training time (Huang et al., 2018; Jia et al., 2017; Ozay & Okatani, 2016). Bansal et al. (Bansal et al., 2018) additionally investigated efficient formulations of the orthogonality constraints. In practice, these constraints are difficult to satisfy, and correspondingly only weakly imposed. In addition, these methods focus on improving performance for a known, given task. This means the training process only extracts features that the network considers useful for improving the performance of the current task, not necessarily improving generalization or transfer performance (Torrey & Shavlik, 2010). While our approach shares similarities with SVD-based constraints, it can be realized with a very efficient $L^2$-based formulation, and takes the full input distribution into account.

Recovering all input information from hidden representations of a network is generally very difficult (Dinh et al., 2016; Mahendran & Vedaldi, 2016), due to the loss of information throughout the layer transformations. In this context, (Tishby & Zaslavsky, 2015) proposed the information bottleneck principle, which states that for an optimal representation, information unrelated to the current task is omitted. This highlights the common specialization of conventional training approaches.

Reversed network architectures were proposed in previous work (Ardizzone et al., 2018; Jacobsen et al., 2018; Gomez et al., 2017), but mainly focus on how to make a network fully invertible via augmenting the network with special structures. As a consequence, the path from input to output is different from the reverse path that translates output to input. Besides, the augmented structures of these approaches can be challenging to apply to general network architectures. In contrast, our approach fully preserves an existing architecture for the backward path, and does not require any operations that were not part of the source network. As such, it can easily be applied in new settings, e.g., adversarial training (Goodfellow et al., 2014). While methods using reverse connections were previously proposed (Zhang et al., 2018a; Teng & Choromanska, 2019), these modules primarily focus on transferring information between layers for a given task, and on auto-encoder structures for domain adaptation, respectively.

## A.2 PRETRAINING AND SINGULAR VALUE DECOMPOSITION

In this section we give a more detailed derivation of our loss formulation, extending Section 3 of the main paper. As explained there, our loss formulation aims for minimizing

$$\mathcal{L}_{\text{RR}} = \sum_{m=1}^{n} (M_m^T M_m \mathbf{d}_m^i - \mathbf{d}_m^i)^2, \tag{11}$$

where $M_m \in \mathbb{R}^{s_m^{\text{out}} \times s_m^{\text{in}}}$ denotes the weight matrix of layer $m$, and data from the input data set $\mathcal{D}_m$ is denoted by $\mathbf{d}_m^i \subset \mathbb{R}^{s_m^{\text{in}}}, i = 1, 2, ..., t$. Here t denotes the number of samples in the input data set.

Minimizing equation 11 is mathematically equivalent to

$$M_m^T M_m \mathbf{d}_m^i - \mathbf{d}_m^i = \mathbf{0} \tag{12}$$

for all $\mathbf{d}_m^i$. Hence, perfectly fulfilling equation 11 would require all $\mathbf{d}_m^i$ to be eigenvectors of $M_m^T M_m$ with corresponding eigenvalues being 1. As in Sec. 3 of the main paper, we make use of an auxiliary orthonormal basis $\mathcal{B}_m : \langle \mathbf{w}_m^1, ..., \mathbf{w}_m^q \rangle$, for which $q$ (with $q \leq t$) denotes the number of linearly independent entries in $\mathcal{D}_m$. While $\mathcal{B}_m$ never has to be explicitly constructed for our method, it can, e.g., be obtained via Gram-Schmidt. The matrix consisting of the vectors in $\mathcal{B}_m$ is denoted by $D_m$.

Since the $\mathbf{w}_m^h \, (h = 1, 2, ...q)$ necessarily can be expressed as linear combinations of $\mathbf{d}_m^i$, equation 11 similarly requires $\mathbf{w}_m^h$ to be eigenvectors of $M_m^T M_m$ with corresponding eigenvalues being 1, i.e.:

$$M_m^T M_m \mathbf{w}_m^h - \mathbf{w}_m^h = \mathbf{0} \tag{13}$$

We denote the vector of coefficients to express $\mathbf{d}_m^i$ via $D_m$ with $\mathbf{c}_m^i$, i.e. $\mathbf{d}_m^i = D_m \mathbf{c}_m^i$. Then equation 12 can be rewritten as:

$$M_m^T M_m D_m \mathbf{c}_m^i - D_m \mathbf{c}_m^i = \mathbf{0} \tag{14}$$

Via an SVD of the matrix $M_m$ in equation 14 we obtain

$$
\begin{aligned}
& M_m^T M_m D_m \mathbf{c}_m - D_m \mathbf{c}_m \\
&= \sum_{h=1}^{q} M_m^T M_m \mathbf{w}_m^h \mathbf{c}_{m_h} - \mathbf{w}_m^h \mathbf{c}_{m_h} \\
&= \sum_{h=1}^{q} V_m \Sigma_m^T \Sigma_m V_m^T \mathbf{w}_m^h \mathbf{c}_{m_h} - \mathbf{w}_m^h \mathbf{c}_{m_h}
\end{aligned}
\tag{15}
$$

where the coefficient vector $\mathbf{c}_m$ is accumulated over the training data set size $t$ via $\mathbf{c}_m = \sum_{i=1}^{t} \mathbf{c}_m^i$. Here we assume that over the course of a typical training run eventually every single datum in $\mathcal{D}_m$ will contribute to $\mathcal{L}_{\mathrm{RR}_m}$. This form of the loss highlights that minimizing $\mathcal{L}_{\mathrm{RR}}$ requires an alignment of $V_m \Sigma_m^T \Sigma_m V_m^T \mathbf{w}_m^h \mathbf{c}_{m_h}$ and $\mathbf{w}_m^h \mathbf{c}_{m_h}$.

By construction, $\Sigma_m$ contains the square roots of the eigenvalues of $M_m^T M_m$ as its diagonal entries. The matrix has rank $r = rank(M_m^T M_m)$, and since all eigenvalues are required to be 1 by equation 13, the multiplication with $\Sigma_m$ in equation 15 effectively performs a selection of $r$ column vectors from $V_m$. Hence, we can focus on the interaction between the basis vectors $\mathbf{w}_m$ and the $r$ active column vectors of $V_m$:

$$
\begin{aligned}
& V_m \Sigma_m^T \Sigma_m V_m^T \mathbf{w}_m^h \mathbf{c}_{m_h} - \mathbf{w}_m^h \mathbf{c}_{m_h} \\
&= \mathbf{c}_{m_h} (V_m \Sigma_m^T \Sigma_m V_m^T \mathbf{w}_m^h - \mathbf{w}_m^h) \\
&= \mathbf{c}_{m_h} \left( \sum_{f=1}^{r} (\mathbf{v}_m^f)^T \mathbf{w}_m^h \mathbf{v}_m^f - \mathbf{w}_m^h \right).
\end{aligned}
\tag{16}
$$

As $V_m$ is obtained via an SVD it contains $r$ orthogonal eigenvectors of $M_m^T M_m$. equation 13 requires $\mathbf{w}_m^1, ..., \mathbf{w}_m^q$ to be eigenvectors of $M_m^T M_m$, but since typically the dimension of the input data set is much larger than the dimension of the weight matrix, i.e. $r \leq q$, in practice only $r$ vectors from $\mathcal{B}_m$ can fulfill equation 13. This means the vectors $\mathbf{v}_m^1, ..., \mathbf{v}_m^r$ in $V_m$ are a subset of the orthonormal basis vectors $\mathcal{B}_m : \langle \mathbf{w}_m^1, ..., \mathbf{w}_m^q \rangle$ with $(\mathbf{w}_m^h)^2 = 1$. Then for any $\mathbf{w}_m^h$ we have

$$
\begin{cases}
(\mathbf{v}_m^f)^T \mathbf{w}_m^h = 1, & \text{if } \mathbf{v}_m^f = \mathbf{w}_m^h \\
(\mathbf{v}_m^f)^T \mathbf{w}_m^h = 0, & \text{otherwise.}
\end{cases}
\tag{17}
$$

Thus if $V_m$ contains $\mathbf{w}_m^h$, we have

$$\sum_{f=1}^{r} (\mathbf{v}_m^f)^T \mathbf{w}_m^h \mathbf{v}_m^f = \mathbf{w}_m^h, \tag{18}$$

and we trivially fulfill the constraint

$$\mathbf{c}_{m_h}(\sum_{f=1}^{r}(\mathbf{v}_m^f)^T\mathbf{w}_m^h\mathbf{v}_m^f - \mathbf{w}_m^h) = \mathbf{0}. \tag{19}$$

However, due to $r$ being smaller than $q$ in practice, $V_m$ typically can not include all vectors from $\mathcal{B}_m$. Thus, if $V_m$ does not contain $\mathbf{w}_m^h$, we have $(\mathbf{v}_m^f)^T\mathbf{w}_m^h = 0$ for every vector $\mathbf{v}_m^f$ in $V_m$, which means

$$\sum_{f=1}^{r}(\mathbf{v}_m^f)^T\mathbf{w}_m^h\mathbf{v}_m^f = \mathbf{0}. \tag{20}$$

As a consequence, the constraint equation 12 is only partially fulfilled:

$$\mathbf{c}_{m_h}(\sum_{f=1}^{r}(\mathbf{v}_m^f)^T\mathbf{w}_m^h\mathbf{v}_m^f - \mathbf{w}_m^h) = -\mathbf{c}_{m_h}\mathbf{w}_m^h . \tag{21}$$

As the $\mathbf{w}_m^h$ have unit length, the factors $\mathbf{c}_m$ determine the contribution of a datum to the overall loss. A feature $\mathbf{w}_m^h$ that appears multiple times in the input data will have a correspondingly larger factor in $\mathbf{c}_m$ and hence will more strongly contribute to $\mathcal{L}_{\text{RR}}$. The $L^2$ formulation of equation 11 leads to the largest contributors being minimized most strongly, and hence the repeating features of the data, i.e., dominant features, need to be represented in $V_m$ to minimize the loss. Interestingly, this argumentation holds when additional loss terms are present, e.g., a loss term for classification. In such a case, the factors $\mathbf{c}_m$ will be skewed towards those components that fulfill the additional loss terms, i.e. favor basis vectors $\mathbf{w}_m^h$ that contain information for about the loss terms. This, e.g., leads to clear digit structures being embedded in the weight matrices for the MNIST example below.

In summary, to minimize $\mathcal{L}_{\text{RR}}$, $V_m$ is driven towards containing $r$ orthogonal vectors $\mathbf{w}_m^h$ which represent the most frequent features of the input data, i.e. the dominant features. It is worth emphasizing that above $\mathcal{B}_m$ is only an auxiliary basis, i.e., the derivation does not depend on any particular choice of $\mathcal{B}_m$.

## A.3 EXAMPLES OF NETWORK ARCHITECTURES WITH PRETRAINING

While the proposed pretraining is significantly more easy to integrate into training pipelines than classic autoencoder pretraining, there are subtleties w.r.t. the order of the operations in the reverse pass that we clarify with examples in the following sections. To specify NN architectures, we use the following notation: $C(k,l,q)$, and $D(k,l,q)$ denote convolutional and deconvolutional operations, respectively, while fully connected layers are denoted with $F(l)$, where $k$, $l$, $q$ denote kernel size, output channels and stride size, respectively. The bias of a CNN layer is denoted with $b$. $I/O(z)$ denote $input/output$, their dimensionality is given by $z$. $I_r$ denotes the input of the reverse pass network. $tanh$, $relu$, $lrelu$ denote hyperbolic tangent, ReLU, and leaky ReLU activation functions (AF), where we typically use a leaky tangent of 0.2 for the negative half-space. $UP$, $MP$ and $BN$ denote $2\times$ nearest-neighbor up-sampling, max pooling with $2 \times 2$ filters and stride 2, and batch normalization, respectively.

Below we provide additional examples how to realize the pretraining loss $\mathcal{L}_{\text{rr}}$ in a neural network architecture. As explained in the main document, the constraint equation 11 is formulated via

$$\mathcal{L}_{\text{rr}} = \sum_{m=1}^{n} \lambda_m \left\| \mathbf{d}_m - \mathbf{d}_m^{'} \right\|_F^2, \tag{22}$$

with $\mathbf{d}_m$, and $\lambda_m$ denoting the vector of activated intermediate data in layer $m$ from the forward pass, and a scaling factor, respectively. $\mathbf{d}_m^{'}$ denotes the activations of layer $m$ from the reverse pass. E.g., let $L_m()$ denote the operations of a layer $m$ in the foward pass, and $L_m^{'}()$ the corresponding operations for the reverse pass. Then $\mathbf{d}_{m+1} = L_m(\mathbf{d}_m)$, and $\mathbf{d}_m^{'} = L_m^{'}(\mathbf{d}_{m+1}^{'})$.

When equation 22 is minimized, we obtain activated intermediate content during the reverse pass that reconstructs the values computed in the forward pass, i.e. $\mathbf{d}_{m+1}^{'} = \mathbf{d}_{m+1}$ holds. Then $\mathbf{d}_m^{'}$ can be reconstructed from the incoming activations from the reverse pass, i.e., $\mathbf{d}_{m+1}^{'}$, or from the output of layer $m$, i.e., $\mathbf{d}_{m+1}$. Using $\mathbf{d}_{m+1}^{'}$ results in a global coupling of input and output throughout

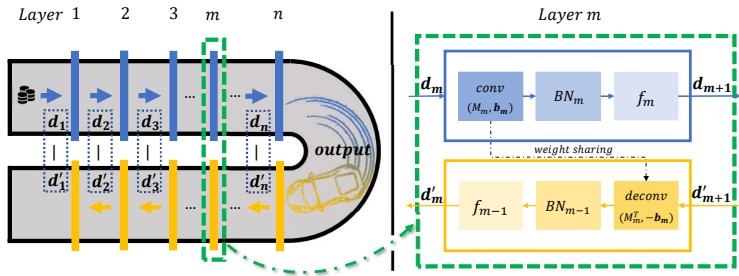

Figure 10: Left: An overview of the regular forward pass (blue) and the corresponding reverse pass (yellow). The right side illustrates how parameters are reused for a convolutional layer. $conv/deconv$ denote convolution/deconvolutional operations. $f_m$ and $BN_m$ denote the activation function and batch normalization of layer $m$, respectively. Shared kernel and bias are represented by $M_m$ and $\mathbf{b}_m$.

all layers, i.e., the *full* loss variant. On the other hand, $\mathbf{d}_{m+1}$ yields a variant that ensures local reversibility of each layer, and yields a very similar performance, as we will demonstrate below. We employ this *local* loss for networks without a unique, i.e., bijective, connection between two layers. Intuitively, when inputs cannot be reliably reconstructed from outputs.

**Full Network Pretraining:** An illustration of a CNN structure with AF and BN and a full loss is shown in figure 10. While the construction of the reverse pass is straight-forward for all standard operations, i.e., fully connected layers, convolutions, pooling, etc., slight adjustments are necessary for AF and BN. It is crucial for our formulation that $\mathbf{d}_m$ and $\mathbf{d}'_m$ contain the same latent space content in terms of range and dimensionality, such that they can be compared in the loss. Hence, we use the BN parameters and the AF of layer $m-1$ from the forward pass for layer $m$ in the reverse pass. An example is shown in figure 14.

To illustrate this setup, we consider an example network employing convolutions with mixed AFs, BN, and MP. Let the network receives a field of $32^2$ scalar values as input. From this input, 20, 40, and 60 feature maps are extracted in the first three layers. Besides, the kernel sizes are decreased from $5 \times 5$ to $3 \times 3$. To clarify the structure, we use ReLU activation for the first convolution, while the second one uses a hyperbolic tangent, and the third one a sigmoid function. With the notation outlined above, the first three layers of the network are

$$
\begin{aligned}
I(32, 32, 1) = \mathbf{d}_1 &\to C_1(5, 20, 1) + \mathbf{b}_1 \to BN_1 \to \text{relu} \\
&\to \mathbf{d}_2 \to MP \to C_2(4, 40, 1) + \mathbf{b}_2 \to BN_2 \to \text{tanh} \\
&\to \mathbf{d}_3 \to MP \to C_3(3, 60, 1) + \mathbf{b}_3 \to BN_3 \to \text{sigm} \\
&\to \mathbf{d}_4 \to ...
\end{aligned}
\tag{23}
$$

The reverse pass for evaluating the loss re-uses all weights of the forward pass and ensures that all intermediate vectors of activations, $\mathbf{d}_m$ and $\mathbf{d}'_m$, have the same size and content in terms of normalization and non-linearity. We always consider states after activation for $\mathcal{L}_{\text{rr}}$. Thus, $\mathbf{d}_m$ denotes activations before pooling in the forward pass and $\mathbf{d}'_m$ contains data after up-sampling in the reverse pass, in order to ensure matching dimensionality. Thus, the last three layers of the reverse network for computing $\mathcal{L}_{\text{rr}}$ take the form:

$$
\begin{aligned}
... &\to \mathbf{d}'_4 \to -\mathbf{b}_3 \to D_3(3, 40, 1) \to BN_2 \to \text{tanh} \to UP \\
&\to \mathbf{d}'_3 \to -\mathbf{b}_2 \to D_2(4, 20, 1) \to BN_1 \to \text{relu} \to UP \\
&\to \mathbf{d}'_2 \to -\mathbf{b}_1 \to D_1(5, 3, 1) \\
&\to \mathbf{d}'_1 = O(32, 32, 1).
\end{aligned}
\tag{24}
$$

Here, the de-convolutions $D_x$ in the reverse network share weights with $C_x$ in the forward network. I.e., the $4 \times 4 \times 20 \times 40$ weight matrix of $C_2$ is reused in its transposed form as a $4 \times 4 \times 40 \times 20$ matrix in $D_2$. Additionally, it becomes apparent that AF and BN of layer 3 from the forward pass do not appear in the listing of the three last layers of the reverse pass. This is caused by the fact that both are required to establish the latent space of the fourth layer. Instead, $\mathbf{d}_3$ in our example represents the activations after the second layer (with $BN_2$ and $tanh$), and hence the reverse pass for $\mathbf{d}'_3$ reuses both functions. This ensures that $\mathbf{d}_m$ and $\mathbf{d}'_m$ contain the same latent space content in terms of range and dimensionality, and can be compared in equation 22.

For the reverse pass, we additionally found it beneficial to employ an AF for the very last layer if the output space has suitable content. E.g., for inputs in the form of RGB data we employ an additional activation with a ReLU function for the output to ensure the network generates only positive values.

**Localized Pretraining:** In the example above, we use a full pretraining with $\mathbf{d}'_{m+1}$ to reconstruct the activations $\mathbf{d}'_m$. The full structure establishes a slightly stronger relationship among the loss terms of different layers, and allows earlier layers to decrease the accumulated loss of later layers. However, if the architecture of the original network makes use of operations between layers that are not bijective, we instead use the local loss. E.g., this happens for residual connections with an addition or non-invertible pooling operations such as max-pooling. In the former, we cannot uniquely determine the $b, c$ in $a = b + c$ given only $a$. And unless special care is taken (Bruna et al., 2013), the source neuron of an output is not known for regular max-pooling operations. Note that our loss formulation has no problems with irreversible operations within a layer, e.g., most convolutional or fully-connected layers typically are not fully invertible. In all these cases the loss will drive the network towards a state that is as-invertible-as-possible for the given input data set. However, this requires a reliable vector of target activations in order to apply the constraints. If the *connection* betweeen layers is not bijective, we cannot reconstruct this target for the constraints, as in the examples given above.

In such cases, we regard every layer as an individual unit to which we apply the constraints by building a localized reverse pass. For example, given a simple convolutional architecture with

$$\mathbf{d}_1 \rightarrow C_1(5, 20, 1) + \mathbf{b}_1 = \mathbf{d}_2 \tag{25}$$

in the forward pass, we calculate $\mathbf{d}'_1$ with

$$(\mathbf{d}_2 - \mathbf{b}_1) \rightarrow D_1(5, 3, 1) = \mathbf{d}'_1, \tag{26}$$

We, e.g., use this local loss in the Resnet110 network below. It is important to note that despite being closer to regular autoencoder pretraining, this formulation still incorporates all non-linearities of the original network structure, and jointly trains full networks while taking into account the original learning objective.

## A.4 MNIST AND PEAK TESTS

Below we give details for the *peak* tests from Sec. 3 of the main paper and show additional tests with the MNIST data set.

*Peak* **Test:** For the *Peak* test we generated a data set of 110 images shown in figure 12. 55 images contain a peak located in the upper left corner of the image. The other 55 contain a peak located in the bottom right corner. We added random scribbles in the images to complicate the task. All 110 images were labeled with a one-hot encoding of the two possible positions of the peak. We use 100 images as training data set, and the remaining 10 for testing. All peak models are trained for 5000 epochs with a learning rate of $0.0001$, with $\lambda = 1e - 6$ for $RR_A$. To draw reliable conclusions, we show results for five repeated runs here. The neural network in this case contains one fully connected layer, with BN and ReLU activation. The results are shown in figure 13, with both peak modes being consistently embedded into the weight matrix of $RR_A$, while regular and orthogonal training show primarily random singular vectors.

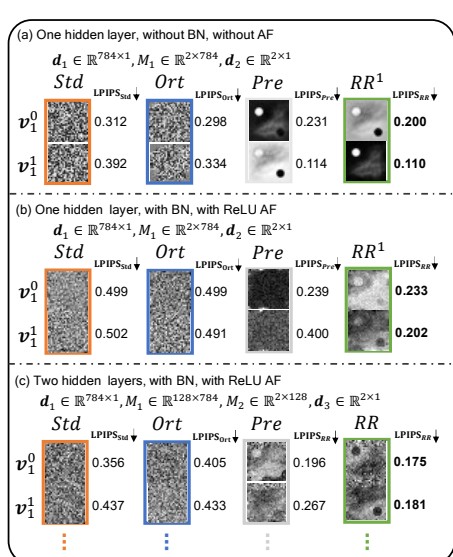

Figure 14: Right singular vectors of $M_1$ for peak tests with different network architectures. Across the three architectures, $RR_A$ successfully extracts dominant and salient features.

We also use different network architectures in figure 14 to verify that the dominant features are successfully extracted when using more complex

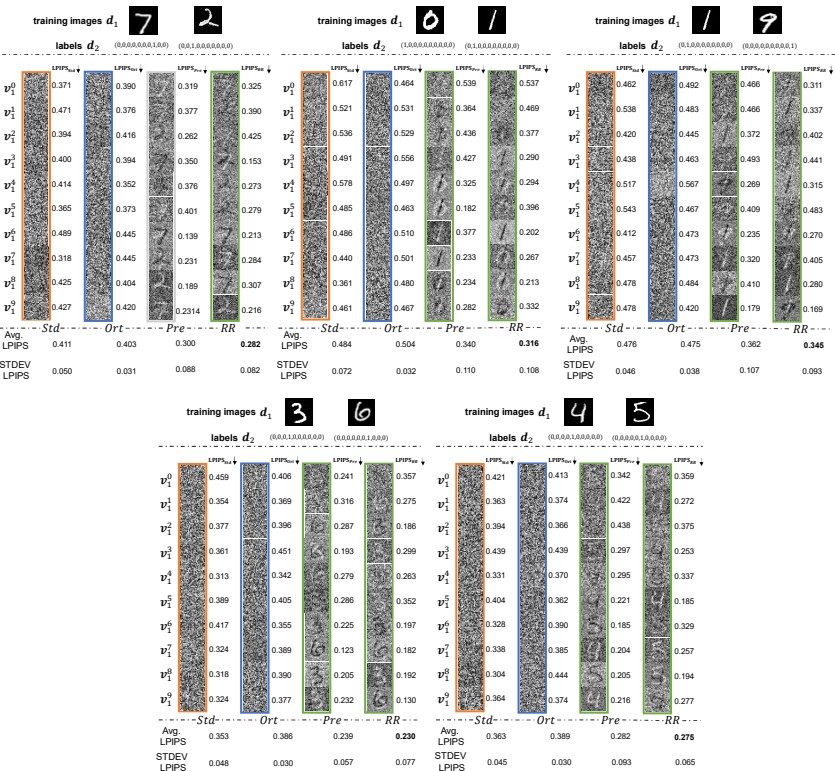

Figure 11: SVD of the $M_1$ matrix for five tests with random two digit images as training data. LPIPS distances (Zhang et al., 2018b) of RR are consistently lower than Std and Ort.

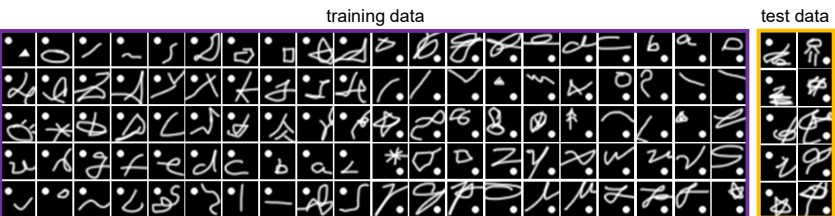

Figure 12: Data set used for the *peak* tests.

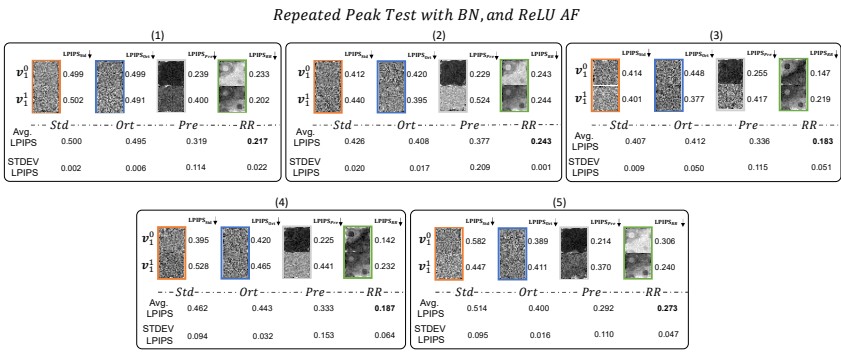

Figure 13: Five repeated tests with the peak data shown in Sec. 3 of the main paper. $RR_A$ robustly extracts dominant features from the data set. The two singular vectors strongly resemble the two peak modes of the training data. This is confirmed by the LPIPS measurements.

network structures. Even for two layers with BN and ReLU activations, our pretraining clearly extracts the two modes of the training data. The visual resemblance is slightly reduced in this case, as the network has the freedom to embed the features in both layers. Across all three cases (for which we performed 5 runs each), our pretraining clearly outperforms regular training and the orthogonality constraint in terms of extracting and embedding the dominant structures of the training data set in the weight matrix.

***MNIST* Test:** We additionally verify that the column vectors of $V_m$ of models from RR training contain the dominant features of the input with MNIST tests, which employ a single fully connected layer, i.e. $\mathbf{d}_2 = M_1 \mathbf{d}_1$. In the first MNIST test, the training data consists only of 2 different images. All MNIST models are trained for 1000 epochs with a learning rate of $0.0001$, and $\lambda = 1e - 5$ for RR$_A$. After training, we compute the SVD for $M_1$. SVDs of the weight matrices of trained models can be seen in figure 11. The LPIPS scores show that features embedded in the weights of RR are consistently closer to the training data set than all other methods, i.e., regular training Std, classic autoencoder pretraining Pre, and regularization via orthogonalization Ort. While the vectors of Std and Ort contain no recognizable structures.

Overall, our experiments confirm the motivation of our pretraining formulation. They additionally show that employing an SVD of the network weights after our pretraining yields a simple and convenient method to give humans intuition about the features learned by a network.

## B MUTUAL INFORMATION

This section gives details of the mutual information and disentangled representation tests from Sec. 4 of the main paper.

### B.1 MUTUAL INFORMATION TEST

Mutual information (MI) measures the dependence of two random variables, i.e., higher MI means that there is more shared information between two parameters. More formally, the mutual information $I(X; Y)$ of random variables $X$ and $Y$ measures how different the joint distribution of $X$ and $Y$ is w.r.t. the product of their marginal distributions, i.e., the Kullback-Leibler divergence $I(X; Y) = \mathrm{KL}[P_{(X,Y)} || P_X P_Y]$, where KL denotes the Kullback-Leibler divergence. Let $I(X; \mathcal{D}_m)$ denote the mutual information between the activations of a layer $\mathcal{D}_m$ and input X. Similarly $I(\mathcal{D}_m; Y)$ denotes the MI between layer $m$ and the output $Y$. We use *MI planes* in the main paper, which show $I(X; \mathcal{D}_m)$ and $I(\mathcal{D}_m; Y)$ in a 2D graph for the activations of each layer $\mathcal{D}_m$ of a network after training. This visualizes how much information about input and output distribution is retained at each layer, and how these relationships change within the network. For regular training, the information bottleneck principle (Tishby & Zaslavsky, 2015) states that early layers contain more information about the input, i.e., show high values for $I(X; \mathcal{D}_m)$ and $I(\mathcal{D}_m; Y)$. Hence in the MI plane visualizations, these layers are often visible at the top-right corner. Later layers typically share a large amount of information with the output after training, i.e. show large $I(\mathcal{D}_m; Y)$ values, and correlate less with the input (low $I(X; \mathcal{D}_m)$). Thus, they typically show up in the top-left corner of the MI plane graphs.

**Training Details:** We use the same numerical studies as in (Shwartz-Ziv & Tishby, 2017) as task $A$, i.e. a regular feed-forward neural network with 6 fully-connected layers. The input variable $X$ contains 12 binary digits that represent 12 uniformly distributed points on a 2D sphere. The learning objective is to discover binary decision rules which are invariant under $O(3)$ rotations of the sphere. $X$ has 4096 different patterns, which are divided into 64 disjoint orbits of the rotation group, forming a minimal sufficient partition for spherically symmetric rules (Kazhdan et al., 2003). To generate the input-output distribution $P(X, Y)$, We apply the stochastic rule $p(y = 1|x) = \Psi(f(x) - \theta), (x \in X, y \in Y)$, where $\Psi$ is a standard sigmoidal function $\Psi(u) = 1/(1 + exp(-\gamma u))$, following (Shwartz-Ziv & Tishby, 2017). We then use a spherically symmetric real valued function of the pattern $f(x)$, evaluated through its spherical harmonics power spectrum (Kazhdan et al., 2003), and compare with a threshold $\theta$, which was selected to make $p(y = 1) = \sum_x p(y = 1|x)p(x) \approx 0.5$, with uniform $p(x)$. $\gamma$ is high enough to keep the mutual information $I(X; Y) \approx 0.99$ bits.

For the transfer learning task $B$, we reverse output labels to check whether the model learned specific or generalizing features. E.g., if the output is [0,1] in the original data set, we swap the entries to [1,0]. 80% of the data (3277 data pairs) are used for training and rests (819 data pairs) are used for testing.

For the MI comparison in Fig. 4 of the main paper, we discuss models before and after fine-tuning separately, in order to illustrate the effects of regularization. We include a model with greedy layer-wise pretraining Pre, a regular model $\text{Std}_A$, one with orthogonality constraints $\text{Ort}_A$, and our regular model $\text{RR}_A$, all before fine-tuning. For the model $\text{RR}_A$ all layers are constrained to be recovered in the backward pass. We additionally include the version $\text{RR}_A^1$, i.e. a model trained with only one loss term $\lambda_1 |\mathbf{d}_1 - \mathbf{d}_1'|_2$, which means that only the input is constrained to be recovered. Thus, $\text{RR}_A^1$ represents a simplified version of our approach which receives no constraints that intermediate results of the forward and backward pass should match. For $\text{Ort}_A$, we used the Spectral Restricted Isometry Property (SRIP) regularization (Bansal et al., 2018),

$$\mathcal{L}_{\text{SRIP}} = \beta \sigma(W^T W - I), \tag{27}$$

where $W$ is the kernel, $I$ denotes an identity matrix, and $\beta$ represents the regularization coefficient. $\sigma(W) = sup_{z \in \mathbb{R}^n, z \neq 0} \frac{\|W_z\|}{\|z\|}$ denotes the spectral norm of $W$.

As explained in the main text, all layers of the first stage, i.e. from $\text{RR}_A$, $\text{RR}_A^1$, $\text{Ort}_A$, $\text{Pre}_A$ and $\text{Std}_A$ are reused for training the fine-tuned models without regularization, i.e. $\text{RR}_{AA}$, $\text{RR}_{AA}^1$, $\text{Ort}_{AA}$, $\text{Pre}_{AA}$ and $\text{Std}_{AA}$. Likewise, all layers of the transfer task models $\text{RR}_{AB}$, $\text{RR}_{AB}^1$, $\text{Ort}_{AB}$, $\text{Pre}_{AB}$ and $\text{Std}_{AB}$ are initialized from the models of the first training stage.

**Analysis of Results:** We first compare the version only constraining input and output reconstruction ($\text{RR}_A^1$) and the full loss version $\text{RR}_A$. Fig. 4(b) of the main paper shows that all points of $\text{RR}_A$ are located in a central region of the MI place, which means that all layers successfully encode information about the inputs as well as the outputs. This also indicates that every layer contains a similar amount of information about $X$ and $Y$, and that the path from input to output is similar to the path from output to input. The points of $\text{RR}_A^1$, on the other hand, form a diagonal line. I.e., this network has different amounts of mutual information across its layers, and potentially a very different path for each direction. This difference in behavior is caused by the difference of the constraints in these two versions: $\text{RR}_A^1$ is only constrained to be able to regenerate its input, while the full loss for $\text{RR}_A$ ensures that the network learns features which are beneficial for both directions. This test highlights the importance of the constraints throughout the depth of a network in our formulation. In contrast, the $I(X; \mathcal{D})$ values of later layers for $\text{Std}_A$ and $\text{Ort}_A$ exhibit small values (points near the left side), while $I(\mathcal{D}; Y)$ is high throughout. This indicates that the outputs were successfully encoded and that increasing amounts of information about the inputs are discarded. Hence, more specific features about the given output data-set are learned by $\text{Std}_A$ and $\text{Ort}_A$. This shows that both models are highly specialized for the given task, and potentially perform worse when applied to new tasks. $\text{Pre}_A$ only focuses on decreasing the reconstruction loss, which results in high $I(X; \mathcal{D})$ values for early layers, and low $I(\mathcal{D}; Y)$ values for later layers.

During the fine-tuning phase for task $A$ (i.e. regularizers being disabled), all models focus on the output and maximize $I(\mathcal{D}; Y)$. There are differences in the distributions of the points along the y-axis, i.e., how much MI with the output is retained, as shown in Fig. 4(c) of the main paper. For model $\text{RR}_{AA}$, the $I(\mathcal{D}; Y)$ value is higher than for $\text{Std}_{AA}$, $\text{Ort}_{AA}$, $\text{Pre}_{AA}$ and $\text{RR}_{AA}^1$, which means outputs of $\text{RR}_{AA}$ are more closely related to the outputs, i.e., the ground truth labels for task $A$. Thus, $\text{RR}_{AA}$ outperforms the other variants for the original task.

In the fine-tuning phase for task $B$, $\text{Std}_{AB}$ stands out with very low accuracy in Fig. 5 of the main paper. This model from a regular training run has large difficulties to adapt to the new task. $\text{Pre}_A$ aims at extracting features from inputs and reconstructed them. $\text{Pre}_{AB}$ outperforms $\text{Std}_{AB}$, which means features helpful for task $B$ are extracted by $\text{Pre}_A$, however, it's hard to guide the feature extracting process. Model $\text{Ort}_{AB}$ also performs worse than $\text{Std}_B$. $\text{RR}_{AB}$ shows the best performance in this setting, demonstrating that our loss formulation yielded more generic features, improving the performance for related tasks such as the inverted outputs for $B$.

We also analyze the two variants of our pretraining: the local variant $l\text{RR}_A$ and the full version $\text{RR}_A$ in terms of mutual information. figure 15 shows the MI planes for these two models, also

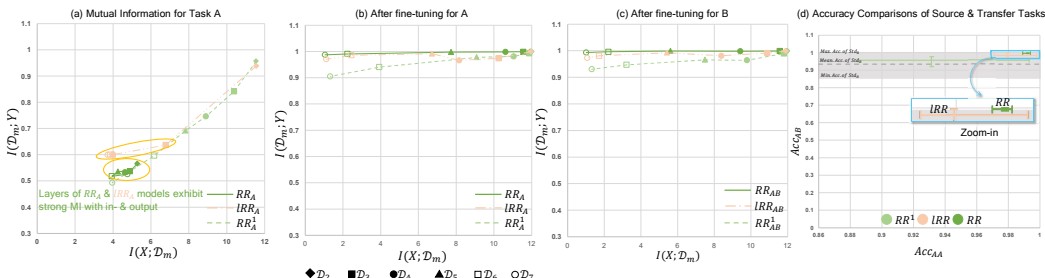

Figure 15: (a-c) MI plane comparisons for local (lRR$_A$) versus full models (RR$_A$). Points on each line correspond to layers of one type of model. a) MI Plane for task A. All points of RR$_A$ and the majority of points for lRR$_A$ (five out seven) are located in the center of the graph, i.e., successfully connect in- and ouput distributions. b,c): After fine-tuning for A/B. The last layer $\mathcal{D}_7$ of RR$_{AA}$ builds the strongest relationship with $Y$. $I(\mathcal{D}_7; Y)$ of lRR$_A$ is only slightly lower than RR$_{AA}$. d): Accuracy comparisons among different models: RR$_{AA}$ yields the highest performance, while lRR$_A$ performs similarly with RR$_{AA}$.

showing RR$_A^1$ for comparison. Despite the local nature of lRR$_A$ it manages to establish MI for the majority of the layers, as indicated by the cluster of layers in the center of the MI plane. Only the first layer moves towards the top right corner, and the second layer is affected slightly. I.e., these layers exhibit a stronger relationship with the distribution of the outputs. Despite this, the overall performance when fine-tuning or for the task transfer remains largely unaffected, e.g., the lRR$_A$ still clearly outperforms RR$_A^1$. This confirms our choice to use the full pretraining when network connectivity permits, and employ the local version in all other cases.

## B.2 DISENTANGLED REPRESENTATIONS

The InfoGAN approach (Chen et al., 2016) demonstrated the possibility to control the output of generative models via maximizing mutual information between outputs and structured latent variables. However, mutual information is very hard to estimate in practice (Walters-Williams & Li, 2009). The previous section and Fig. 4(b) of the main paper demonstrated that models from our pretraining (both RR$_A^1$ and RR$_A$) can increase the mutual information between network inputs and outputs. Intuitively, the pretraining explicitly constrains the model to recover an input given an output, which directly translates into an increase of mutual information between input and output distributions compared to regular training runs. For highlighting how our pretraining can yield disentangled representations (as discussed in the later paragraphs of Sec. 4 of the main text), we follow the experimental setup of InfoGAN (Chen et al., 2016): the input dimension of our network is 74, containing 1 ten-dimensional category code $c_1$, 2 continuous latent codes $c_2, c_3 \sim \mathcal{U}(-1, 1)$ and 62 noise variables. Here, $\mathcal{U}$ denotes a uniform distribution.

**Training Details:** As InfoGAN focuses on structuring latent variables and thus only increases the mutual information between latent variables and the output, we also focus the pretraining on the corresponding latent variables. I.e., the goal is to maximize their mutual information with the output of the generative model. Hence, we train a model RR$^1$ for which only latent dimensions $c_1, c_2, c_3$ of the input layer are involved in the loss. We still employ a full reverse pass structure in the neural network architecture. $c_1$ is a ten-dimensional category code, which is used for controlling the output digit category, while $c_2$ and $c_3$ are continuous latent codes, to represent (previously unknown) key properties of the digits, such as orientation or thickness. Building relationship between $c_1$ and outputs is more difficult than for $c_2$ or $c_3$, since the 10 different digit outputs need to be encoded in a sinlge continuous variable $c_1$. Thus, for the corresponding loss term for $c_1$ we use a slightly larger $\lambda$ factor (by 33%) than for $c_2$ and $c_3$. Details of our results are shown in figure 16. Models are trained using a GAN loss (Goodfellow et al., 2014) as the loss function for the outputs.

**Analysis of Results:** In figure 16 we show additional results for the disentangling test case. It is visible that our pretraining of the RR$^1$ model yields distinct and meaningful latent space dimensions for $c_{1,2,3}$. While $c_1$ controls the digit, $c_{2,3}$ control the style and orientation of the digits. For comparison, a regular training run with model Std does result in meaningful or visible changes when adjusting the latent space dimensions. This illustrates how strongly the pretraining can shape the

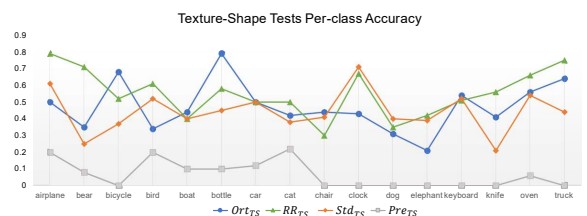

(a) Varying $c_1$ for Std (No clear meaning)    (b) Varying $c_1$ for $RR^1$ (Digit type)    (c) Varying $c_2$ for $RR^1$ (Style tweaking)    (d) Varying $c_3$ for $RR^1$ (Rotation)

Figure 16: Additional results for the disentangled representations with the MNIST data: For every row in the figures, we vary the corresponding latent code (left to right), while keeping all other inputs constant. Different rows indicate a different random noise input. For example, in (b): every column contains five results which are generated with different noise samples, but the same latent codes $c_{1\sim3}$. In every row, 10 results are generated with 10 different values of $c_1$, which correspond to one digit each for (b). (a): For a regular training (Std), no clear correspondence between $c_1$ and the outputs are apparent (similarly for $c_{2,3}$). (c): Different $c_2$ values result in a tweaked style, while $c_3$ controls the orientation of the digit, as shown in (d). Thus, in contrast to Std, the pretrained model learns a meaningful, disentangled representation.

latent space, and in addition to an intuitive embedding of dominant features, yield a disentangled representation.

## C   DETAILS OF EXPERIMENTAL RESULTS

### C.1   TEXTURE-SHAPE BENCHMARK

**Training Details:** All training data of the texture-shape tests were obtained from (Geirhos et al., 2018). The stylized data set contains 1280 images, 1120 images are used as training data, and 160 as test data. Both edge and filled data sets contain 160 images each, all of which are used for testing only. All three sets (stylized, edge, and filled) contain data for 16 different classes.

Texture-Shape Tests Per-class Accuracy

$\rightarrow Ort_{TS}$   $\rightarrow RR_{TS}$   $\rightarrow Std_{TS}$   $\rightarrow Pre_{TS}$

Figure 17: Separate per-class test accuracies for the four model variants. The $RR_{TS}$ model exhibits a consistently high accuracy across all 16 classes.

**Analysis of Results:** For a detailed comparison, we list per-class accuracy of stylized data training runs for $Ort_{TS}$, $Std_{TS}$, $Pre_{TS}$ and $RR_{TS}$ in figure 17. $RR_{TS}$ outperforms the other three models for most of the classes. $RR_{TS}$ requires an additional $41.86\%$ for training compared to $Std_{TS}$, but yields a $23.76\%$ higher performance. (Training times for these models are given in the supplementary document.) All models saturated, i.e. training $Std_{TS}$ or $Ort_{TS}$ longer does not increase classification accuracy any further. We also investigated how much we can reduce model size when using our pretraining in comparison to the baselines. A reduced model only uses $67.94\%$ of the parameters, while still outperforming $Ort_{TS}$.

### C.2   GENERATIVE ADVERSARIAL MODELS

**Training Details:** The data set of smoke simulation was generated with a Navier-Stokes solver from an open-source library (Thuerey & Pfaff, 2018). We generated 20 randomized simulations with 120 frames each, with 10% of the data being used for training. The low-resolution data were down-sampled from the high-resolution data by a factor of 4. Data augmentation, such as flipping and rotation was used in addition. As outlined in the main text, we consider building an autoencoder model for the synthetic data as task $B_1$, and a generating samples from a real-world smoke data set as task $B_2$. The smoke capture data set for $B_2$ contains 2500 smoke images from the ScalarFlow data set (Eckert et al., 2019), and we again used 10% of these images as training data set.

Task $A$: We use a fully convolutional CNN-based architecture for generator and discriminator networks. Note that the inputs of the discriminator contain high resolution data $(64, 64, 1)$, as well as low resolution $(16, 16, 1)$, which is up-sampled to $(64, 64, 1)$ and concatenated with the high resolution data. In line with previous work (Xie et al., 2018), $RR_A$ and $Std_A$ are trained with a

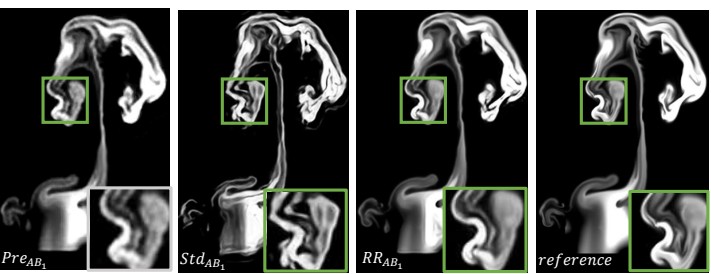

Figure 18: Example outputs for $\text{Pre}_{AB_1}$, $\text{Std}_{AB_1}$, $\text{RR}_{AB_1}$. The reference is shown for comparison. $\text{RR}_{AB_1}$ produces higher quality results than $\text{Std}_{AB_1}$ and $\text{Pre}_{AB_1}$.

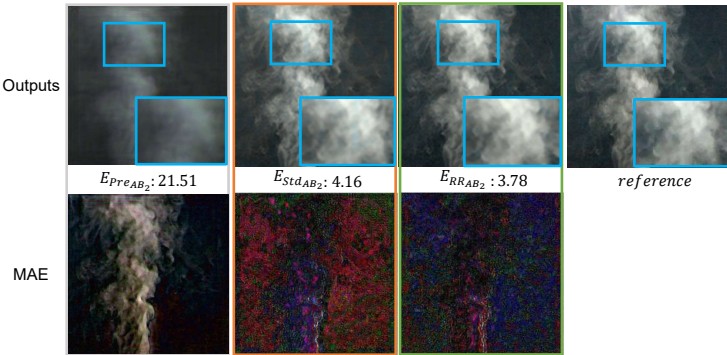

Figure 19: Mean Absolute Error (MAE) comparisons for smoke task $B_2$ models. $\text{RR}_{AB_2}$ shows the smallest error, and additionally achieves the best visual quality amongst the different models.

non-saturating GAN loss, feature space loss and L2 loss as base loss function. All generator layers are involved in the pretraining loss. As greedy layer-wise autoencoder pretraining is not compatible with adversarial training, we pretrain $\text{Pre}_A$ for reconstructing the high resolution data instead.

Task $B_1$: All encoder layers are initialized from $\text{RR}_A$ and $\text{Std}_A$ when training $\text{RR}_{AB_1}$ and $\text{Std}_{AB_1}$. It is worth noting that the reverse pass of the generator is also constrained when training $\text{Pre}_A$ and $\text{RR}_A$. So both encoder and decoder are initialized with parameters from $\text{Pre}_A$ and $\text{RR}_A$ when training $\text{Pre}_{AB_1}$ and $\text{RR}_{AB_1}$, respectively. This is not possible for a regular network like $\text{Std}_{AB_1}$, as the weights obtained with a normal training run are not suitable to be transposed. Hence, the deconvolutions of $\text{Std}_{AB_1}$ are initialized randomly.

Task $B_2$: As the data set for the task $B_2$ is substantially different and contains RBG images (instead of single channel gray-scale images), we choose the following setups for the $\text{RR}_A$, $\text{Pre}_A$ and $\text{Std}_A$ models: parameters from all six layers of $\text{Std}_A$ and $\text{RR}_A$ are reused for initializing decoder part of $\text{Std}_{AB_2}$ and $\text{RR}_{AB_2}$, parameters from all six layers of $\text{Pre}_A$ are reused for initializing the encoder part of $\text{Pre}_{AB_2}$. Specially, when initializing the last layer of $\text{Pre}_{AB_2}$, $\text{Std}_{AB_2}$ and $\text{RR}_{AB_2}$, we copy and stack the parameters from the last layer of $\text{Pre}_A$, $\text{Std}_A$ and $\text{RR}_A$, respectively, into three channels to match the dimenions of the outputs for task $B_2$. Here, the encoder part of $\text{RR}_{AB_2}$ and the decoder of $\text{Pre}_{AB_2}$ are not initialized with $\text{RR}_A$ and $\text{Pre}_A$, due to the significant gap between training data sets of task $B_1$ and task $B_2$. Our experiments show that only initializing the decoder part of $\text{RR}_{AB_2}$ (avg. loss:$1.56e7$, std. dev.:$3.81e5$) outperforms initializing both encoder and decoder (avg. loss:$1.82e7 \pm 2.07e6$), and only initializing the encoder part of $\text{Pre}_{AB_2}$ (avg. loss:$4.41e7 \pm 6.36e6$) outperforms initializing both encoder and decoder (avg. loss:$9.42e7 \pm 6.11e7$). We believe the reason is that initializing both encoder and decoder part makes it more difficult to adjust the parameters for new data set that is very different from the data set of the source task.

**Analysis of Results:** Example outputs of $\text{Pre}_{AB_1}$, $\text{Std}_{AB_1}$ and $\text{RR}_{AB_1}$ are shown in figure 18. It is clearly visible that $\text{RR}_{AB_1}$ gives the best performance among these models. We similarly illustrate the behavior of the transfer learning task $B_2$ for images of real-world fluids. This example

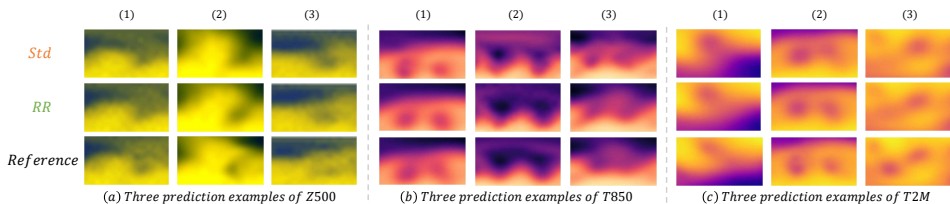

*(a) Three prediction examples of Z500*    *(b) Three prediction examples of T850*    *(c) Three prediction examples of T2M*

Figure 20: A comparison of additional Z500, T850, T2M predictions (zoomed in regions). The predictions inferred by the RR model are closer to the observed references.

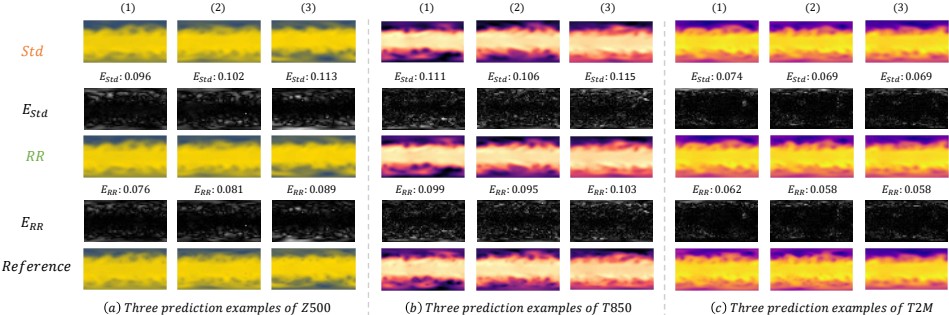

*(a) Three prediction examples of Z500*    *(b) Three prediction examples of T850*    *(c) Three prediction examples of T2M*

Figure 21: MSE value comparisons between RR and Std($E_{RR}$ for RR and $E_{Std}$ for Std). RR consistently yields lower errors than Std.

likewise uses an autoencoder structure. Visual comparisons are provided in figure 19, where $RR_{AB_2}$ generates results that are closer to the reference. Overall, these results demonstrate the benefits of our pretraining for GANs, and indicate its potential to obtain more generic features from synthetic data sets that can be used for tasks involving real-world data.

## C.3 WEATHER FORECASTING

**Training Details:**  The weather forecasting scenario discussed in the main text follows the methodology of the *WeatherBench* benchmark (Rasp et al., 2020). This benchmark contains 40 years of data from the ERA5 reanalysis project Hersbach et al. (2020) which was re-sampled to a 5.625° resolution, yielding $32 \times 64$ grid points in ca. two-hour intervals. Data from the year of 1979 to 2015 (i.e., 162114 samples) are used for training, the year of 2016 for validation. The last two years (2017 and 2018) are used as test data. All RMSE measurements are latitude-weighted to account for area distortions from the spherical projection.

The neural networks for the forecasting tasks employ a ResNet architecture with 19 layers, all of which contain 128 features with $3 \times 3$ kernels (apart from $7 \times 7$ in the first layer). All layers use batch normalization, leaky ReLU activation (tangent 0.3), and dropout with strength 0.1. As inputs, the model receives feature-wise concatenated data from the WeatherBench data for 3 consecutive time steps, i.e., $t$, $t - 6h$, and $t - 12h$, yielding 117 channels in total. The last convolution jointly generates all three output fields, i.e., pressure at 500 hPa (Z500), temperature at 850 hPa (T850), and the 2-meter temperature (T2M).

**Analysis of Results:**  In addition to the quantitative results for both years of test data given in the main text, figure 20 and 21 contain additional example visualizations from the test data set. A visualization of the spatial error distribution w.r.t. ground truth result in also shown in figure 21. It becomes apparent that our pretraining achieves reduced errors across the whole range of samples. Both temperature targets contain a larger number of smaller scale features than the pressure fields. While the gains from our pretraining approach are not huge (on the order of 3% in both cases), they represent important steps forward. The learning objective is highly non-trivial, and the improvements were achieved with the same limited set of training data. Being very easy to integrate into existing training pipelines, these results indicate that the proposed pretraining methodology has the potential to yield improved learning results for a wide range of problem settings.

