# OpenReview forum: "Reviving Autoencoder Pretraining"
_ICLR.cc/2021/Conference — Reject_

### Official Review · AnonReviewer4 · 2020-10-27
**Lack of novelty**

**Rating:** 4
**Confidence:** 4

**Review:**

This paper proposes to use orthogonal weight constraints for autoencoders. The authors demonstrate that under orthogonal weights (hence invertible), more features could be extracted. The theory is conducted under linear cases while the authors claim it can be applied to more complicated scenarios such as higher dimension and with nonlinearity. The experiments demonstrate the performance of proposed model on classification tasks and generative tasks. Several baselines are compared.

The paper is poorly written. It is full of inconsistent and irrelevant claims. The method is not clarified. All experiments are in low quality.

+ves:

+ This paper discusses its connection to several of topics such as mutual information, greedy learning, SVD etc.


Concerns:

- No novelty. Using orthogonal weight regularization has been widely studied.

- The theory does not apply to higher dimensional cases or nonlinear cases. The discussion seems trivial. There are no connection of corresponding theory and the model in the experiments.

- Throughout the paper, the "pretraining" process is not clarified.

- The authors claims applying their method to GAN but I don't see how they combine their model with it.

- The classification and generation experiment results are not convincing. In Figure 6, 7, the difference are in range of error bar. In Figure 8, 9, there is no advantage from proposed method.

=====POST-REBUTTAL COMMENTS========

I would like to thank the authors for their response. The authors have clarified the method in their response. I appreciate all the experimental details in the appendix. I tend to agree this is a promising idea and worth explored.

However, this paper clearly cannot be accepted in its current form. The paper is poorly organized and poorly written. The method in the paper needs to be clarified. Their theory goes nowhere and proves nothing. In terms of the experimental results, the authors choose unclear baselines (which they claim state-of-the-art) and report improvement in terms of percentage increase (percentage over percentage). None of this is convincing to me.

I slightly increased my score.

---

> ### Author Response · Authors · 2020-11-16
> **Reply to reviewer #4**
>
> Dear reviewer,
>
> Thank you for taking the time to review our ICLR submission. We have to admit that we were surprised about the generic and short criticism, and the resulting assessment. We believe there are several fundamental misunderstandings which we try to address below. Please also use this forum to let us know whether we have clarified the points of the review. We’d be happy to provide further details here.
>
> Q1: “No novelty. Using orthogonal weight regularization has been widely studied.”
>
> A1: This is a first important misunderstanding - our paper does not propose “yet-another-orthogonalization”. We chose  orthogonality constraints for our derivation, since the formulation of our proposed method shares similarities with orthogonality constraints, and we were hoping this would provide a good intuition for readers. While the changes we make seem small in the equations, their influence is significant and leads to an algorithm that has little in common with orthogonal regularizers.
>
> We bring input and output data sets into the formulation, which yields orthogonal “features”, rather than just enforcing weight orthogonality as orthogonality constraints. Compared to these methods, our approach builds very different relationships between input, network intermediate layers, and output (Figure 4(b)). Also, results in Figure 3, 11, 13, 14 verify that our methods extract much more generic and reusable features from the data set than models trained with orthogonality constraints.
>
>
> Q2: “The theory does not apply to higher dimensional cases or nonlinear cases. The discussion seems trivial. There are no connection of corresponding theory and the model in the experiments.”
>
> A2: For the derivation in Section 3, we used a single layer as an example, and we do not apply any assumptions regarding input and output of the neural network layer. Hence our derivation applies to networks with more layers and higher dimensional cases. While we keep the derivation for nonlinear activation functions and batch normalization as future work, we include a large number of experiments that show our method works in these cases. For instance, the Resnet-110 contains ca. 1.7 million trainable parameters for CIFAR-10, the generator network with 0.87 million trainable parameters for super resolution, and the models for weather forecasting with 6.36 million trainable parameters. All models in Section 3 and Section 4 are trained with nonlinear activation functions often make use of batch normalization.
>
> Q3: “Throughout the paper, the "pretraining" process is not clarified.”
>
> A3: We follow standard procedure. Like in numerous previous works, models trained with constraints (i.e. our proposed loss / orthogonal constraints / auto-encoder layers) are what we refer to as pre-training models. Afterwards, the constraints are removed, and the model is fine tuned.
>
> Q4: “No experimental detail is given, e.g. dataset, model architecture, training procedure, evaluation metric etc.”
>
> A4: We are very surprised about this comment: All experimental details, such as dataset, model architectures, training parameters and etc., are discussed in our supplementary document. Could you please be more specific which information was actually omitted?
>
> Q5: “The authors claim applying their method to GAN but I don't see how they combine their model with it.”
>
> A5: In subsection “Generative Adversarial Models” of Section 4, we applied our proposed approach to GAN training. Our method is easy to integrate via a reverse pass of the forward network and the L2 constraints for intermediate layers. Details of the reverse pass architecture are given in Table 14.
>
> Q6: “The classification and generation experiment results are not convincing. In Figure 6, 7, the difference are in range of error bar. In Figure 8, 9, there is no advantage from proposed method.”
>
> A6: Again, we think there was a severe misunderstanding here: our method outperforms the others across all of the shown test cases in terms of mean performance. Following common practice, the graphs show mean and standard deviation across several models due to the inherent randomness of training. Hence, in Figure 6 our RR model has an improvement that is on average 16% better than a regular training, while both AE and orthogonality deteriorate. For a state-of-the-art CIFAR 10 model, our RR training has yielded almost 75% increase in average performance over orthogonality. We’d be curious to hear from the reviewer where our results do not make the robust improvements in performance clear.

---

### Official Review · AnonReviewer2 · 2020-10-29
**Review of Reviving Autoencoder Pretraining.**

**Rating:** 3
**Confidence:** 3

**Review:**

The authors present an approach to pre-training of an ANN which utilizes a purportedly novel approach.  This approach aims to be "data aware" by incorporating the individual data points in the orthogonality constraint in the loss function.

The paper is well-written and his mathematically rigorous.  The mathematics appears to be correct.  It is a concept paper, which I appreciate.

The paper puts the proposed approach in juxtaposition with the greedy, layer-by-layer approach.

The authors show improved performance on a handful of tasks against some baseline tasks.  The baselines were, mostly, well selected with a major exception (see below).

The authors provide some experimental data analysis showing that the features learned by the hidden layers in the proposed approach are more easily interpretable by humans.  Furthermore, the features learned by the baseline features appear to be noisy whereas the features learned by the proposed approach capture the structure seen in the data.  Finally, the authors have some interesting results indicating that the features learned by this approach lead to greater generalization or transferability between tasks/data sets.

My largest critique of this work is the lack of discussion/comparison with the long-standing auto-encoder approaches which can already compute PCA/SVD/orthogonal basis vectors utilizing a single hidden layer.  Are these methods unrelated in some way?  If so, why?  This work should be tied back to these approaches.

Another critique of this work is the lack of future work directions.  The authors offered no questions about the proposed approach which were generated during their line of inquiry.  No analysis of where the proposed approach might be inferior to existing approaches was provided or discussed.

---

> ### Author Response · Authors · 2020-11-16
> **Reply to reviewer #2**
>
> Dear reviewer,
>
> Thank you very much for your detailed review. We are glad to hear that mathematical derivations of our proposed method were considered to be rigorous.
>
> Q1: “My largest critique of this work is the lack of discussion/comparison with the long-standing auto-encoder approaches which can already compute PCA/SVD/orthogonal basis vectors utilizing a single hidden layer. Are these methods unrelated in some way?”
>
> A1: PCA and auto-encoders are typically used for dimensionality reduction. The goal of our approach differs, as we aim for improving the performance of neural networks. Thus it’s not our main goal to extract SVD or orthogonal basis vectors. We primarily use the SVDs to illustrate which features from the data set are embedded into our models, both visually and in our derivation, but our goal is to improve the actual performance of a trained model rather than the set of singular vectors.
>
> We also choose auto-encoder pretraining to motivate our method, since its forward pass and reverse pass network structures share similarities with auto-encoder structures. However, the theory behind our approach is very different to auto-encoder pretraining, and shares more similarities with orthogonalization.
>
> We build a reverse pass network by reusing weights from the forward pass network, and constrain the input layer and intermediate layers to reconstruct features from the inputs. Besides, constraints in the output layer guide the network to extract features for an original task. Auto-encoder pretraining also pushes the network to reconstruct the input, however, features extracted by the network are random and uncontrolled. Hence, they cannot be easily reused in future tasks.
>
> Besides, auto-encoder pretraining is not applicable for more complicated (yet commonly used) networks, such as networks with skip connections. Our approach does not have such problems (the weather forecasting uses skip connections), and we even employ with GANs (cf. Section 4). Hence, we focused on showing improvements in terms of performance in these settings, rather than comparisons to methods for PCA / SVD extraction.
>
> Q2: “Another critique of this work is the lack of future work directions. The authors offered no questions about the proposed approach which were generated during their line of inquiry.”
>
> A2: As future work we point out that we do not yet include activation functions and batch normalization in our derivation, though our experimental results show that our approach works consistently for networks with/without activation functions or batch normalization. As for additional limitations, our formulation results in a moderate increase in computational cost, which is right now discussed on page 7 in the 2nd paragraph.
>
> Unfortunately, due to space restrictions, we kept the outlook very brief and only mentioned time predictions and explainability / interpretability there. However, throughout our tests the pretraining did not fare worse, but rather gave improvements for the vast majority of cases. So we think there are many applications that will benefit directly from our approach. We will illustrate both future work and limitations clearer in a  revised version.

---

### Official Review · AnonReviewer1 · 2020-11-05
**An interesting and well described approach for unsupervised pretraining of neural networks**

**Rating:** 9
**Confidence:** 3

**Review:**

Quality

The paper is well written and mathematically precise. Statements are supported through helpful illustrations and the appendix provides the reader with sufficient information about the experimental setup and evaluations.

Clarity

The approach is clearly motivated and is described in full mathematical detail. The authors also show a highly efficient way to integrate the actually computational very expensive loss into neural networks. The derivation is moderately easy to follow for readers with a background in linear algebra.

Originality

The proposed is, while inspired by previous orthogonalization approaches, a novel idea and its relation to previous work is discussed appropriately.

Significance

Unsupervised pretraining in itself has a large significance for deep learning, even though it lost in popularity due to other approaches that achieved a similar result without the extra preparation phase of the neural network training. It is important and useful to keep the research in this area alive and the authors contributed very valuable knowledge with this paper.

Pros:

-- very promising approach for pretraining of neural networks

-- well written paper with good illustrations

-- large experimental evaluation

Cons

-- derivation only for networks without activation function

-- evaluation very much focused on image tasks

-- Figure 5: legend hardly readable (kind of true for many figures)

-- source code not referenced (might be due to remain anonymous during review)

---

> ### Author Response · Authors · 2020-11-16
> **Reply to reviewer #1**
>
> Dear reviewer,
>
> Thank you very much for your review and comments. We are glad to read the positive assessment of our proposed method.
>
> Q1: “Derivation only for networks without activation function”
>
> A1: Currently our derivation focuses on networks without activation function and batch normalization. However, our experiments (peak tests in figure 3 and all experiments in Section 4) show that our approach works vert consistently for networks with activation functions and batch normalization. We see the inclusion of non-linear components such as activation functions or batch normalization into the derivation as a very interesting direction for  future work.
>
> Q2: “Evaluation is very much focused on image tasks”
>
> A2: Our experiments show that our pre-training is beneficial for all kinds of tasks that we have encountered during our tests, and we tried to give a broad overview in our paper (given the restrictions for space and attention of readers). Hence, we first selected the basic mutual information tests, then continued with different image tasks as they represent widely used baselines, and then moved to tasks related to physics (fluids & weather). We will definitely apply our approach to new tasks as future work.
>
> Q3: “Source code not referenced (might be due to remain anonymous during review)”
>
> A3: Due to anonymity, we have not referenced our source code repository in our submission, but we commit to publishing the full source code and data upon acceptance.

---

### Official Review · AnonReviewer5 · 2020-11-06
**A good work that uses a simple method to improve the generalising capabilities, but need to be improved.**

**Rating:** 5
**Confidence:** 4

**Review:**

This paper proposes an auto-encoder pre-training approach for regularising the neural network parameters, which can be used in many different existing neural models. The proposed approach is build based on the unsupervised auto-encoder pre-training and the orthogonality constraints. A number of classical applications are shown to be improved using the proposed models.



Strengths of the paper:

1. The proposed approach is a general approach, easy to integrate into the existing models.

2. The experimental results demonstrate the gains in accuracy for original and new tasks below for a wide range of applications



Weaknesses of the paper:

1. The writing of the paper needs to be improved. Many typos and informal/improper places can be easily found, e.g

   - Left double quotation marks in latex should be $``$, rather than $”$.

   - Paragraph before Eq (9), "equation 10 " should be "equation 8 ", since equation 10 has no $d_{m+1}{′}$.

   - $L^2$ and $L2$, $L_{rr}$ and $L_{RR}$, are confused in this paper.
   - CNNs, Autoencoders, and GANs, although are well-known, should cite with the corresponding references.

2. The proposed approach is very simple, which is incremented from (Bansal et al., 2018), hence the contribution is limited.

3. The motivation of the proposed models but not utilize other models for the problem is not clear.

4. My main concern is the organisation of this paper, where the current version is not suitable for general readers in different domains. The preliminary knowledge of the proposed model is unclear. I see at least two key points need to be discussed at the beginning of the paper, i.e. regularisers, and unsupervised auto-encoder pertaining, however, are not well illustrated.

5. The model section should be organised with subsections, where the differences of the preliminary models and the proposed approaches should be explicitly indicated.

6. What is missing when doing the orthogonality constraint, comparing with other regularisation approaches, should be discussed.

7. Comparing the proposed $L_{RR}$ (Eq. (3)) with Eq. (1) (Bansal et al., 2018), the only difference is the weighting/regularizing of parameters by the input data $d_{m}^{I}$. However, the input space could be arbitrarily in value scales, Eq. (3) could punish the large/sharp values of the input space, which may not be desirable for some tasks where the input spaces are very sparse and discrete.

Overall, this is good work that uses a simple method to improve the generalising capabilities of many models, but there is a number of weaknesses as indicated above.

*************
Updates: Thanks for the authors' response. However, not of all my concerns are well-addressed. Based on the current methodology and the novelty of this paper, I remain my overall rating of this paper unchanged.

---

> ### Author Response · Authors · 2020-11-16
> **Reply to reviewer #5**
>
> Dear reviewer,
>
> Thank you so much for your review and suggestions. We are glad that you consider this paper as good work, which improves generalising model capacity via simple approach. Below, we will address your questions in more detail.
>
> Q1: “The proposed approach is very simple, which is incremented from (Bansal et al., 2018), hence the contribution is limited.”
>
> A1: Our proposed method actually significantly differs from orthogonality constraints or auto-encoder pretraining. We chose the orthogonality constraint for our derivation, since the formula form of our proposed method looks similar. However, the influence of our change is significant. We bring input and output data sets into the formulation, which pushes the network to extract orthogonal features from the data set, rather than just enforcing weight orthogonality as orthogonality constraints. This seemingly small change leads to a very implementation, and different relationships between input, network intermediate layers, and output (see Figure 4(b)). Results in Figure 3, 11, 13, 14 also verify that our methods extracts much better features from the data set than models trained with orthogonality constraints.
>
>
> Q2: “The motivation of the proposed models but not utilize other models for the problem is not clear.”
>
> A2: The structure of our networks is influenced by invertible network architectures. But we aim for learning a general representation as pre-training, we constrain the network to an as-reversible-as-possible process for all intermediate layer activations.
>
> Q3: “My main concern is the organisation of this paper, where the current version is not suitable for general readers in different domains.”
>
> A3: Thank you for your feedback. Because of the page limitations, the additional discussion of related work had to be moved to Appendix Section A.1. To improve the organization in the revised version, we’d be happy to add more discussion about regularizers and unsupervised auto-encoder pretraining earlier on in the introduction section. More specifically, drawbacks of orthogonality constraints, reasons why unsupervised auto-encoder pretraining was diminished will be discussed.
>
> Q4: “What is missing when doing the orthogonality constraint, comparing with other regularisation approaches, should be discussed.”
>
> A4: For orthogonal constraint, we chose the state-of-the-art orthogonal constraint, Spectral RestrictedIsometry Property (SRIP) regularization  (Bansal et al., 2018). In texture-shape tests (Figure 6), our proposed method (RR_TS) outperforms the SRIP model (Ort_TS) by 16.38% for the stylized data set, 16.75% for the edge data set and 9.68% for the filled data set. In CIFAR 10 classification tests (Figure 7), the orthogonality constraint brings 0.4% accuracy improvements for the state-of-the-art model, but 0.7% from our proposed model.
>
> Q5: “Comparing the proposed L_RR  (Eq. (3)) with Eq. (1) (Bansal et al., 2018), the only difference is the weighting/regularizing of parameters by the input data d_{m}^{I}. However, the input space could be arbitrarily in value scales, Eq. (3) could punish the large/sharp values of the input space, which may not be desirable for some tasks where the input spaces are very sparse and discrete.”
>
> A5: Our method represents a pre-training approach, which aims at extracting dominant features from the dataset, so that our model can be reused in related tasks and improve model performance. On the other hand, in this pre-training stage, our proposed approach will focus less on specialized or sharp features, but focus on more generic and dominant features (as illustrated in the derivation in Section 3). Our method has no problems with “sharp” changes, though. E.g., in the MNIST digit controlling test (Figure 1(b) and Figure 16), input code c_1 used to control the output digit category. It is a ten-dimensional category code, which is sparse and discrete. Our proposed method still can successfully build strong relationships between input code c_1 and the output, which is consistent with other tests, such as Figure 4(b).

---

### Author Response · Authors · 2020-11-16
**General response to reviewers**

We would like to thank all the reviewers for their insightful comments and suggestions. The comments about unclear parts and potential weaknesses are very important for us to refine and improve our paper.

One point that we believe is worth highlighting here is that the realization of our method fundamentally differs from typical orthogonality constraints: we employ reversible layers, more akin to autoencoder pretraining. At the same time, our method also substantially differs from regular autoencoder pretraining by employing full passes over deep networks, in conjunction with additional losses and nonlinearities. Among others, it is much more generic.

We also substantially outperform both: a state-of-the-art regularization with orthogonality is worse by 19.2%, while autoencoder pretraining only achieves 38% of our performance (both for the texture-shape tests of Figure 6). Hence our method yields significant improvements for practically relevant scenarios.

We thank the reviewers for the comments regarding misunderstandings, and we will clarify differences to orthogonality and autoencoder pretraining in future revisions.

Below, we address questions of the individual reviews in more detail.

---

### Decision · Program_Chairs · 2021-01-07
**Final Decision**

**Decision:**

Reject

**Comment:**

This work presents a practical unsupervised pretraining strategy that does not require layer-wise training stages. Clearly this is an area that has lot of potential and the work seems to head in the right direction.
However, despite a very positive review, I share the same concerns raised by the remaining 3 reviewers. Better motivation and clarity is needed and, considering the proposed approach, a much more thorough comparison and analysis of the theoretical advantages and guarantees of such an approach.
A main argument is that the method can handle arbitrary networks due to the way it is implemented, it is however not clear how practical that would be and how that would work in presence of non-linearities.
Experiments require also additional work to be presented with clear and standard baselines, not by presenting SOTA on arbitrary tasks. This seemed to be a main concern of the reviewers.